# InfiBench: Evaluating the Question-Answering Capabilities of Code Large Language Models

**Linyi Li**
Simon Fraser University
`linyi_li@sfu.ca`

**Shijie Geng**
ByteDance Inc & Rutgers University
`sg1309@rutgers.edu`

**Zhenwen Li**[*]  **Yibo He**[*]  **Hao Yu**[*]  **Ziyue Hua**[*]
Peking University

**Guanghan Ning**
ByteDance Inc

**Siwei Wang**
ByteDance Inc

**Tao Xie**
Key Lab of HCST (PKU), MOE
`taoxie@pku.edu.cn`

**Hongxia Yang**
The Hong Kong Polytechnic University (PolyU)
`hongxia.yang@polyu.edu.hk`

## Abstract

Large Language Models for code (code LLMs) have witnessed tremendous progress in recent years. With the rapid development of code LLMs, many popular evaluation benchmarks, such as HumanEval, DS-1000, and MBPP, have emerged to measure the performance of code LLMs with a particular focus on code generation tasks. However, they are insufficient to cover the full range of expected capabilities of code LLMs, which span beyond code generation to answering diverse coding-related questions. To fill this gap, we propose **InfiBench**, the **first large-scale freeform question-answering (QA) benchmark for code** to our knowledge, comprising 234 carefully selected high-quality Stack Overflow questions that span across 15 programming languages. InfiBench uses four types of model-free automatic metrics to evaluate response correctness where domain experts carefully concretize the criterion for each question. We conduct a systematic evaluation for over 100 latest code LLMs on InfiBench, leading to a series of novel and insightful findings. Our detailed analyses showcase potential directions for further advancement of code LLMs. InfiBench is fully open source at `https://infi-coder.github.io/infibench` and continuously expanding to foster more scientific and systematic practices for code LLM evaluation.

## 1 Introduction

In recent years, Large Language Models (LLMs) have been revolutionizing the software development landscape [17, 12], demonstrating exceedingly strong and comprehensive capabilities in comprehending, generating, debugging, and summarizing code [9, 24]. For example, code LLM-powered products like GitHub Copilot [14] reached millions of active users within just one year of their launch.

Alongside the huge success of proprietary LLMs such as GPT-3.5 / GPT-4 [36] and Gemini [13], the development of open-source code LLMs[2] [35, 43, 39, 30] has been advancing at an unprecedented fast pace. As of June 2024, the Hugging Face Open LLM Leaderboard [4] has cataloged over 3,300 submissions of such models.

---

[*]Equal contribution.

[2]We define code LLMs as LLMs that show decent capabilities in the code domain, no matter whether they are exclusively trained or finetuned with code data or not.

38th Conference on Neural Information Processing Systems (NeurIPS 2024) Track on Datasets and Benchmarks.

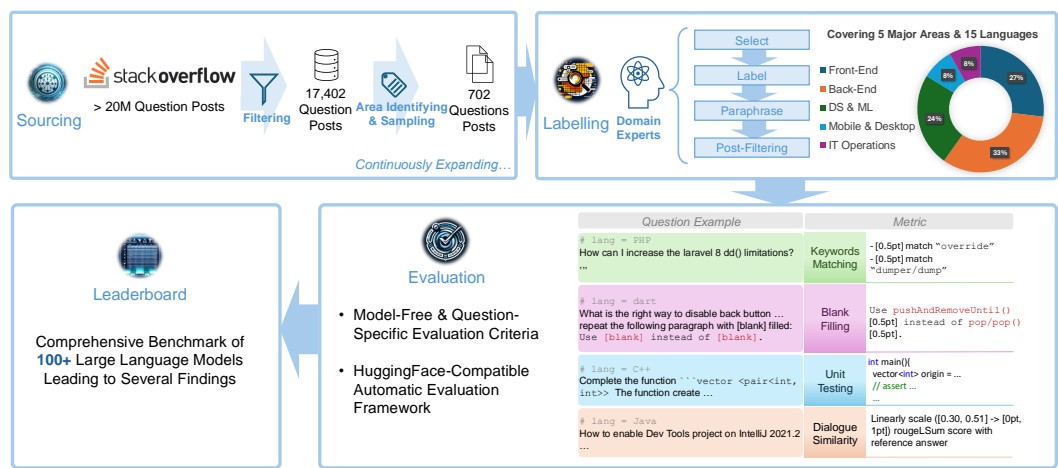

Figure 1: **InfiBench overview**. We construct the InfiBench benchmark by filtering high-quality and diverse question posts from Stack Overflow and annotating question-level evaluation criteria with domain experts. With an model-free automatic evaluation framework, we evaluate over 100 latest code LLMs (one of the most extensive evaluations for code LLMs to the best of our knowledge), leading to several insightful findings.

Table 1: **Comparison between InfiBench and common existing benchmarks.** Existing benchmarks weigh heavily on code generation, unit-test-based evaluation, and major programming languages. InfiBench processes a much higher diversity to reflect real-world code LLMs' usage scenarios. More discussion in Section 2.6.

| Benchmark | Domain | # Question | Evaluation | Data Source | Highest LLM Score |
|---|---|---|---|---|---|
| HumanEval [9] | Python Programming | 164 | Test Cases | Hand-Written | 90.2% |
| MBPP [3] | Python Programming | 974 | Test Cases | Hand-Written | 81.1% |
| APPS [16] | Python Programming | 10,000 | Test Cases | Competitions | / (no report yet) |
| DS-1000 [21] | Python Programming | 1,000 | Test Cases + Surface Form Constraints | StackOverflow | / (no report yet) |
| HumanEval+ [26] | Python Programming | 164 | Augmented Test Cases | HumanEval | 86.6% |
| HumanEvalPack [33] | Repair, Explain, Generation in 6 Languages | 2,952 | Test Cases | HumanEval | 47.8%/52.1%/78.3% |
| LBPP [32] | Python Programming | 161 | Test Cases | Hand-Written | 64% |
| SWE-bench [19] | Python Debugging / Repair | 2,294 | Test Cases | GitHub | 22.06% |
| SWE-bench Verified [38] | Python Debugging / Repair | 500 | Test Cases | SWE-bench | 45.20% |
| InfiBench | Free-Form Code Question Answering in 15 Languages | 234 | Keyword + Blank Filling + Test Cases + Text Similarity | Stack Overflow | 70.64% |

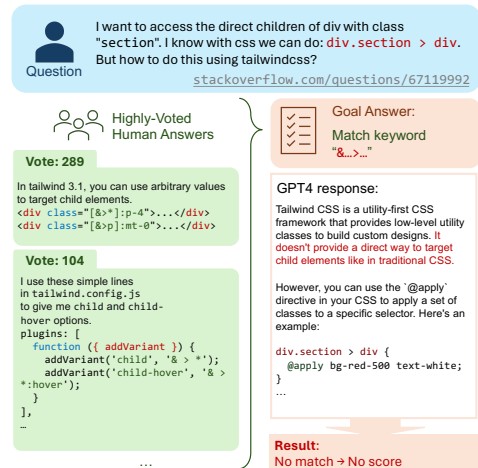

Figure 2: A challenging question paraphrased from Stack Overflow where GPT-4 fails to answer.

Given the plethora of code LLMs available, the development of reliable code benchmarks seems to lag in four aspects: (1) **Benchmarks beyond code generation are relatively few.** Benchmarks for code LLMs typically focus on a specific task or domain, often overly focus on code generation. For example, the widely-used HumanEval [9] and MBPP [3] purely focus on Python code generation, and DS-1000 [21] focuses on Python code generation in the field of data science. (2) **Independent code benchmarks are relatively few.** Recent efforts evolve existing benchmarks (e.g., HumanEval) to include more scenarios [33], languages [48], and tests [26]. However, these efforts lead to a series of benchmarks sharing the same source data (e.g., HumanEval Python problems), reducing score independence. (3) **Existing code benchmarks are saturating.** Strong LLMs are saturating existing benchmarks, e.g., GPT-4 has already achieved 90.2% Pass@1 score on HumanEval [37], while in real-world scenarios, GPT-4 can still fail as exemplified in Figure 2. (4) **Common benchmarks may be contaminated.** Some LLMs have unconventional high performance in common benchmarks and are suspected to have memorized benchmark-related data [10, 46, 32], obscuring the evaluation results. *Can we systematically and comprehensively evaluate code LLMs' abilities in challenging real-world usage scenarios?*

To answer the question, we introduce InfiBench, a systematic benchmark for evaluating the free-form question-answering capabilities of code LLMs. As the first benchmark of its kind, the core principle of InfiBench aims to accurately represent how developers interact with and utilize such models in real-world scenarios. To achieve this, InfiBench comprises 234 questions that are carefully selected and proportionally filtered from the natural high-quality question distribution of Stack Overflow, without any constraints on topics, programming languages, question types, or answer forms. As a result, the curated 234 questions span 15 programming languages and 5 major areas: *front-end*, *back-end*, *DS&ML (data science and machine learning)*, *mobile and desktop*, and *ITOps (information technology operations)*.

Question diversity comes with evaluation challenges for two reasons. (1) Lack of metric. Unlike code generation or multiple-choice benchmarks, which can be evaluated through standardized methods like unit testing, there is no universal metric for response correctness for free-form questions. (2) Challenges with model-based evaluation. Model-based evaluations such as those involving GPT-4 are not only costly but also raise concerns about privacy and bias.

To mitigate the evaluation challenges, InfiBench includes an automatic evaluation framework that integrates four types of *model-free* metrics: keyword matching, blank filling, unit testing, and dialogue similarity. For each question, we invite industry domain experts to paraphrase the prompt, select the most appropriate metric, and write down the concrete criteria using domain-specific knowledge, with highly-voted answers from Stack Overflow as a reference. These questions and evaluation criteria are then cross-validated to ensure correctness and objectiveness and further calibrated to improve consistency across languages. Human experiments show that InfiBench evaluation aligns with humans better than LLM-based evaluation, achieving 85.1% agreement rate compared to 77.8% achieved by GPT-4o-based evaluation.

As a novel and systematic benchmark disjoint with existing ones in terms of both forms and data sources, we believe that InfiBench is an ideal tool to measure existing code LLMs objectively. Hence, we conduct a systematic evaluation for **over 100 code LLMs** spanning both proprietary and open-source worlds using the InfiBench framework — the latest and most extensive evaluation for code LLMs to the best of our knowledge. Our evaluation leads to several insightful findings: (1) On InfiBench, GPT-4 achieves a score of $70.64\%$, being far from perfect but still far exceeding the most capable open-source models as of June 2024. On the other hand, GPT3.5 is surpassed by a few open-source models. (2) At similar model sizes, coding LLMs are usually visibly stronger than general LLMs; finetuning LLMs are usually visibly stronger than base LLMs. (3) The performance differences between different model families are huge, where one model could surpass another with less than 1/10 parameters, highlighting the importance of training data quality and techniques. (4) The scaling law is empirically verified for open-source models with fewer than 40B parameters, but not for those with more, where a turning point emerges. InfiBench is fully open source under CC BY-SA 4.0 license and continuously expanding[3], including both the benchmark and Hugging-Face-compatible evaluation tools. All resources are available at `https://infi-coder.github.io/infibench`.

## 2 Benchmark Creation

InfiBench is created from a high-quality subset of Stack Overflow questions up until June 14, 2023. In this section, we describe the data curation process and the evaluation framework in detail.

### 2.1 Data Curation

Stack Overflow is a question-and-answer website for developers with more than 24 million registered users as of June 2024 [41]. Since the website is a large collection of natural and diverse coding questions from real-world developers, we believe that questions from Stack Overflow can effectively evaluate code LLM's capabilities in real-world usage scenarios.

The full Stack Overflow dataset contains 23.54 million question posts and 34.68 million answer posts. Each question post has a total view count. Each answer post is attached to a question and has a vote count. The question creator can choose one answer as officially accepted.

---

[3]In other words, **infi**nitely expanding, after which the **bench**mark is named.

Table 2: **InfiBench data statistics by area and language**. We uniformly sample a subset from the initial seed set (see Section 2.1) according to the area quota (see Section 2.2) for domain experts to select questions and annotate the correctness criterion to construct the benchmark.

| Area | Language | Initial Seed Set | | Tentative | Final InfiBench Benchmark | | | |
| | | # Questions | % Area Quota | # Questions Quota | # Questions Quota | % Questions Quota | # Area Quota | % Area Quota |
|---|---|---|---|---|---|---|---|---|
| Front-End | Javascript | 4912 | | 44 | 44 | 18.80% | | |
| | CSS | 87 | 40.41% | 10 | 10 | 4.27% | 63 | 26.92% |
| | HTML | 600 | | 10 | 9 | 3.85% | | |
| Back-End | Java | 930 | | 18 | 17 | 7.26% | | |
| | C# | 629 | | 12 | 12 | 5.13% | | |
| | PHP | 462 | | 10 | 9 | 3.85% | | |
| | Go | 117 | 18.71% | 10 | 9 | 3.85% | 77 | 32.91% |
| | Ruby | 71 | | 10 | 10 | 4.27% | | |
| | Rust | 96 | | 10 | 10 | 4.27% | | |
| | C/C++ | 287 | | 10 | 10 | 4.27% | | |
| DS & ML | Python | 2779 | 21.39% | 47 | 47 | 20.09% | 56 | 23.93% |
| | R | 184 | | 10 | 9 | 3.85% | | |
| Mobile & Desktop | Dart | 1562 | | 19 | 19 | 8.12% | 19 | 8.12% |
| | Kotlin | 383 | | 10 | | | | |
| | Swift | 551 | 18.13% | 10 | Removed during Post-Filtering (see Section 2.3) | | | |
| | VBA | 16 | | 9 | | | | |
| IT Ops. | Bash | 188 | 1.36% | 21 | 19 | 8.12% | 19 | 8.12% |
| Total | | 13854 | 100.0% | 270 | 234 | 100.00 % | 234 | 100.00% |

As we aim to create a benchmark where the correctness evaluation criteria are clear, we view the positively voted answers as an important reference source. Hence, we choose to keep only the questions that have at least three positively voted answers and an officially accepted answer, which turn out to be 1,090,238 questions. For these one million questions, we choose to keep questions that are frequently viewed and relatively new. To fulfill this criterion, we draw a scatter plot of these $\approx$ 1 million questions, plotting the number of days since their creation until June 14, 2023 (data collection end-date) on the $x$-axis against the logarithm of their view counts on the $y$-axis. As shown in Figure 3, we empirically determine to keep questions that lie above the line connecting $(0, 5)$ and $(3000, 15.5)$, resulting in a subset of 17,402 questions.

Utilizing the mandatory question tags of these questions, we then manually construct a tag tree that covers the 200 most frequent tags, enabling us to identify the top programming languages and areas for 14,330 out of these 17,402 questions. These questions are from 24 programming languages, with each language being categorized into one primary area among the five (front-end, back-end, DS&ML, mobile and desktop, and ITOps). Lastly, we exclude 6 programming languages that either describe data or are domain-specific: JSON, regex, Markdown, YAML, CSV, and SQL. As a result, we compile 13,854 questions that serve as the *initial seed set*.

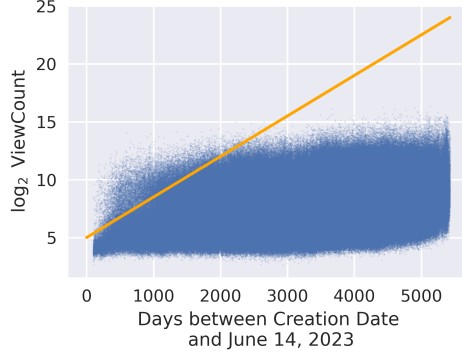

Figure 3: Scatter plot of filtered Stack Overflow questions. Questions above the orange line kept.

## 2.2 Sampling

Based on a user study of developers' demand from our organization, we allocate the tentative area quota to be 25%, 25%, 25%, 15%, and 10% for front-end, back-end, DS&ML, mobile and desktop, and IT Ops, respectively. Inspired by HumanEval size and considering the labelling labor cost, we set 200 questions as the target benchmark size. Hence, the tentative size quotas by area are 50, 50, 50, 30, and 20 respectively. We then proportionally distribute the area quotas to language quotas based on the frequency of each language in the initial seed set. However, we observe that following this rule, certain languages such as CSS and C/C++ end up with fewer than 10 questions, which may yield unreliable language-level sub-score, so, for these languages, we set their quotas to 10.

As a result, we derive the *tentative* question quota for each language as shown in Table 2, which sums up to 270 questions. After determining the tentative question quota, we uniformly sample from the initial seed set a roughly two times larger pool for the domain expects to select and annotate.

## 2.3   Human Annotation

We recruited five domain experts inside our company to create the benchmark, each in charge of one area. The annotation process is composed of three steps:

- **Step 1: Question Selection and Type Annotation.** Domain experts select high-quality questions from the inspecting set and annotate the question type to be one of the four: code completion, code debugging, config and environment debugging, and knowledge question-answering.
- **Step 2: Prompt Paraphrasing.** Domain experts paraphrase and simplify the original question body into succinct and explicit instructions. We include this step for two main purposes: (1) Reduce domain gap. From user-shared conversations collected from ShareGPT, we observe that when interacting with code LLMs, users tend to provide short and direct instructions like "Fix problem..." and "Debug code...". However, when posting Stack Overflow questions, users tend to be lengthy with courtesy words. We ask the domain experts to paraphrase the question to code LLM user's style without changing the semantics. (2) Reduce the impact of memorization and data contamination. Some code LLMs may be trained or finetuned with Stack Overflow data. Paraphrasing the questions can help to mitigate the result advantages of these models. Benchmark results in Table 4 reveal the effectiveness of this step where copying Stack Overflow answers only achieves a 65.18% score. We defer further discussion in Section 2.5.
- **Step 3: Correctness Criterion Annotation.** Domain experts choose one or multiple evaluation metrics from our supported ones (see Section 2.4) and annotate the concrete criterion following a YAML schema. External files can be attached if needed, e.g., unit tests and reference answers.

**Calibration and Post-Filtering.** To improve annotation consistency and objectiveness, we introduce a few checkpoints for domain experts to read others' annotated cases, discuss them, and reach consensus for controversial cases. After the 270 tentative questions were annotated, we then ran an initial evaluation of all these questions on over 30 code LLMs. This initial evaluation helps us to identify questions whose criteria are incorrect or out of distribution. We filter out these questions and then remove all questions from Kotlin, Swift, and VBA languages since the questions in these languages are too few after filtering. After this calibration and post-filtering process, the final benchmark includes 234 questions spanning over 15 languages. Their statistics are shown in Table 2. As we can observe, compared to the population area distribution of high-quality Stack Overview questions (see "% Area Quota" column under "Initial Seed Set"), the area distribution of final benchmark questions (see "% Area Quota" column under "Final InfiBench Benchmark") is more balanced and less biased towards front-end, mobile, and desktop topics.

## 2.4   Evaluation Criteria and Evaluation Framework

In response to the diversified questions, InfiBench evaluation framework integrates four types of model-free and automatic metrics as below. Domain experts choose one or multiple metric types along with their weights and concretize.

- **Keywords Matching.** Though the responses can be in diverse forms, for a significant portion of benchmark questions, we find that the existence of some keywords strongly determines the quality of the response. Domain experts can write rules that match keywords and regular expressions or construct recursive logical expressions on top of keyword-matching results. When multiple keywords exist, each matching result can have its weight in the final score.
- **Blank Filling.** For some questions, it is challenging to measure the correctness given the response uncertainty. In this case, domain experts can instruct the model to answer the question by following a given template and filling in the blanks in the template. The blanks can correspond to either natural language or code snippet. Then, similar to keywords matching, each blank can match potential keywords, regular expressions, or recursive logic expressions built upon matching results. This metric type tests not only the model's QA ability but also its instruction-following ability.
- **Unit Testing.** For code-intensive questions, we can follow existing benchmarks to evaluate response correctness by unit tests. For this type, domain experts may add more specifications in the prompt to allow for unit-test-based evaluation, such as specifications on function name, input arguments, and output format. Domain experts can further import the context setup and cleanup script.
- **Dialogue Similarity.** For natural-language-intensive questions, domain experts can extract and shorten the reference answers from Stack Overflow, and then use the ROUGE score [25] to evaluate the response similarity with reference answers. The ROUGE score was initially proposed and

widely used in evaluating the quality of text summarization and machine translation. To map the ROUGE score back to our benchmark scale, we allow domain experts to tune the mapping interval and scores within the interval are then linearly mapped to our score scale.

The example questions and corresponding criteria are illustrated in Figure 1. Detail statistics of metric type ratios, question type ratios, and prompt length are shown in Table 3.

**Score Computation.** We treat each question equally with one point each. Given 234 questions in the benchmark, the full score is 234, and we by default report the percentage score (achieved score divided by 234) unless otherwise noted. The one point for each question can be further decomposed into a few scoring points within each question. For example, a question may contain four keywords with weights 2, 1, 1, and 1 each. Then, matching each keyword can contribute to 0.4, 0.2, 0.2, and 0.2 points respectively to the final score.

Table 3: InfiBench statistics.

(a) Question type.

| Question Type | Ratio |
|---|---|
| Code Completion | 30.37% |
| Knowledge Question-Answering | 27.04% |
| Code Debugging | 26.67% |
| Config & Environment Debugging | 15.93% |

(b) Metric type.

| Metric Type | Ratio |
|---|---|
| Keywords Matching | 57.41% |
| Blank Filling | 12.22% |
| Unit Testing | 19.26% |
| Dialogue Similarity | 11.85% |

(c) Prompt token length with Code Llama tokenizer.

| min | 25% quantile | median | mean | 75% quantile | max |
|---|---|---|---|---|---|
| 43 | 145.75 | 223 | 338.46 | 359.50 | 5047 |

**Implementation.** We have implemented an automated evaluation framework with Python, publicly available at `https://infi-coder.github.io/infibench`. Specifically, for blank-filling evaluation, we use the longest common subsequence matching via dynamic programming to capture the filled blanks in the response. For unit-testing evaluation, we construct a runtime environment that supports the test execution for nine languages. We plan to integrate the framework into the Hugging Face Open LLM Leaderboard [4] to further ease the evaluation burden.

**How does InfiBench Evaluation Align with Human?** To evaluate the alignment between InfiBench evaluation and human expert evaluation, we randomly sample 100 questions without replacement from the benchmark and select three strong LLMs to generate responses: GPT-4-0613, GPT-3.5-turbo, and Mistral Codestral-22b. For each question, we randomly choose two out of these three model responses to construct response pairs, resulting in 100 response pairs $\mathcal{R} = \{(A_i, B_i) : 1 \leq i \leq 100\}$. For each response pair $(A, B) \in \mathcal{R}$, we use InfiBench, GPT-4o, and human expert to evaluate into four outcomes: $A$ is more correct than $B$ ($A > B$); $B$ is more correct then $A$ ($B > A$); both $A$ and $B$ are correct ($A \approx B \uparrow$); both $A$ and $B$ are incorrect ($A \approx B \downarrow$). *Our purpose is to evaluate how InfiBench evaluation aligns with humans, specifically when compared to the widely-used LLM-as-a-judge (i.e., model-based evaluation) [47].* The concrete grading criteria is as below:

- InfiBench gives a score between $[0, 1]$ for each response in the pair. If the score difference in the pair is larger than 0.2, we label the outcome to be $A > B$ or $B > A$ respectively; otherwise, if the maximum score among the two is larger than 0.5, we label the outcome to be $A \approx B \uparrow$; otherwise, we label the outcome to be $A \approx B \downarrow$.

- For GPT-4o evaluation, we deploy the prompting template from LLM-Blender [18, Appendix E] and trigger GPT-4o for grading the four outcomes. We enhance the reliability of the comparison by switching $A$ and $B$ and prompting GPT-4o twice. We record the preference only when a consistent preference exists.

- For human evaluation, we recruit human annotators who came up with the criteria to label the comparison preference since they are familiar with the questions and have strong expertise. Annotators have no access to the evaluation results of InfiBench and GPT-4o, nor which source model generates the response. Annotators were instructed to directly label each pair with the four outcomes.

We defer the consensus matrices between InfiBench/GPT-4o and human annotators along with more findings in Appendix C. If we only count the cases where both InfiBench/GPT-4o and humans have clear preferences, the agreement rate between InfiBench and humans is 85.1%, and the agreement rate between GPT-4o and humans is 77.8%. Hence, *the InfiBench evaluation aligns with human experts better than the GPT-4o evaluation (with >80% confidence).* We observe that the advantage of InfiBench comes from the ability to detect deceptive answers. some model responses pretend to be helpful with lengthy wording and hallucinations. GPT-4o is more likely to be cheated than InfiBench, which looks for key concepts that should exist in a helpful answer.

### 2.5 Mitigations on Memorization and Data Contamination

InfiBench is created from the publicly available Stack Overflow corpus to reflect real-world scenarios, and this corpus may already exist in the training set of some code LLMs (e.g., DeepSeek Coder [15] and StarCoder 2 [28]). Hence, some code LLMs may achieve a high score simply due to memorization. To mitigate this, we asked the domain experts to paraphrase every question as an essential step (see Section 2.3). Hence, copying either the highly voted answers or officially accepted answers of the original questions only achieves 65.18%, being far from perfect and inferior to GPT-4's 70.64%. Furthermore, code LLMs that use Stack Overflow data do not demonstrate significant advantages over those without. Hence, we deem the effect of contamination as small.

On the other hand, we release the post IDs of the source question posts of InfiBench. Hence, future LLM training could consider this benchmark to conduct deduplication and ablation studies on data contamination. Another usage of our benchmark is to evaluate retrieval-augmented (RAG) code LLMs where perfect retrieval from Stack Overflow and moderate adaptation should solve these questions, which we leave as future work.

### 2.6 Comparison with Existing Benchmarks

In Table 1, we compare InfiBench with several existing benchmarks for code LLMs. As reflected in the table, InfiBench strongly complements existing benchmarks for code LLMs by (1) extending them beyond code generation to a wide range of real-world tasks, (2) diversifying them since InfiBench does not share the same source as existing ones, and (3) increasing the differentiation as an unsaturated benchmark. Related benchmarks are further illustrated in Section 5. On the other hand, the benchmark is limited in size due to the high cost of correctness criteria labelling, and we are continuously expanding the benchmark.

## 3 Evaluation and Leaderboard

We systematically evaluated over 100 code LLMs spanning both proprietary and open-source worlds on InfiBench. To the best of our knowledge, this is the most extensive evaluation for code LLMs.

**Evaluation Protocol.** We adopt best@10 as the main evaluation metric: 10 responses are sampled and evaluated for each question, then the best score per question is recorded and summed up. Throughout the evaluation, we set sampling temperature $T = 0.2$ and top $p = 0.9$.

Furthermore, we swept sampling parameters with GPT-4 and the detailed results are in Appendix G. In a nutshell, for maximizing the performance under best@10, the best parameters are $T = 1.0$ and $p = 0.9$, leading to a score of $76.15\% \pm 0.21\%$ (in comparison to $70.64\% \pm 0.82\%$ in our main setting $T = 0.2, p = 0.9$). In particular, the temperature $T$ affects much and the effect of top $p$ is minor. We decided to stick to the original parameters $T = 0.2$ and $p = 0.9$ in the main evaluation since this setting is more akin to the real-world scenario where user generates once with low temperature.

We design two system prompts (shown in Appendix H), one for normal questions and the other for open-ended questions with an additional sentence to encourage succinct responses. For generic models, we generate the prompt with "`{system prompt}\n{content prompt}`" format; for instruction-finetuned or chat models, we generate the prompt with their prompt templates.

For proprietary models, we evaluate the latest models from OpenAI (GPT-4, GPT-4o, etc), Anthropic (Claude 3), and Mistral AI (Mistral Small/Medium/Large) with API calling. When budget permits, we repeat each evaluation three times and report standard deviation. For open-source models, we download models from Hugging Face and evaluate them on an 8xA100 server with bigcode-evaluation-harness [5]. When the model size is within 30B parameters, we repeat each evaluation three times and report the standard deviation. All raw model responses are available at `https://figshare.com/articles/dataset/InfiBench_Detail_Evaluation_Data/26104864`. More details on the evaluation protocol are in Appendix E.

**Leaderboard.** In Table 4, we present aggregated InfiBench leaderboards by model family, model type, and model size. The full leaderboard is deferred to Appendix E due to space limit. The table includes scores from using the original Stack Overflow answer posts as reference. Results are also presented as a scatter plot in Figure 4, where normal models are shown as scatters with error bars,

Table 4: **Aggregated InfiBench leaderboards (best viewed zoomed in and in color).** "Size" column records number of parameters. For MoE models, "total params. / params. activated during inference" is recorded. Bar colors stand for General Base , General Finetuned , Code Base , and Code Finetuned models respectively. Icon "🔒" stands for proprietary models otherwise open-source. Full leaderboard in Appendix E.

(a) InfiBench leaderboard by model family, where best model within each model family is shown.

| | Family | Best Model Name | Size | InfiBench Score |
|---|---|---|---|---|
| 1 | 🔒GPT-4 | GPT-4-0613 | ? | 70.64% ± 0.82% |
| 2 | DeepSeek Coder | deepSeek-coder-V2-instruct | 236B / 21B | 65.49% |
| 3 | 🔒Claude 3 | Claude 3 Opus | ? | 63.89% |
| 4 | Mistral Open | Codestral-22b | 22B | 62.98% ± 0.56% |
| 5 | Phind | Phind-CodeLlama-34B-v2 | 34B | 59.00% |
| 6 | 🔒Mistral | mistral-large | ? | 58.22% |
| 7 | DeepSeek LLM | deepseek-llm-67b-chat | 67B | 57.41% |
| 8 | 🔒GPT-3.5 | GPT-3.5-turbo-0613 | ? | 56.47% ± 1.34% |
| 9 | Qwen | Qwen-72B | 72B | 55.34% |
| 10 | Magicoder | Magicoder-S-CL-7B | 7B | 52.71% ± 0.72% |
| 11 | WizardLM | WizardCoder-Python-34B-V1.0 | 34B | 52.59% |
| 12 | Code Llama | CodeLlama-34b-Instruct | 34B | 50.45% |
| 13 | 01.AI | Yi-34B-Chat | 34B | 49.58% |
| 14 | Zephyr | Zephyr 7B beta | 7B | 46.31% ± 1.11% |
| 15 | StarCoder2 | 15B-Instruct | 15B | 45.89% ± 0.95% |
| 16 | DeepSeek MoE | deepseek-moe-16b-chat | 16B / 2.8B | 45.18% ± 1.65% |
| 17 | OctoPack | OctoCoder | 15.5B | 44.55% ± 0.79% |
| 18 | gemma | gemma-7b-it | 7B | 40.68% ± 1.23% |
| 19 | Llama 2 | Llama2-70B-Chat | 70B | 39.30% |
| 20 | InternLM | InternLM-Chat-20B | 20B | 37.41% ± 0.75% |
| 21 | Baichuan2 | Baichuan2-13B-Chat | 13B | 34.40% ± 1.34% |
| 22 | StarCoder | StarCode+ | 15.5B | 30.67% ± 1.57% |
| 23 | CodeGen2.5 | CodeGen2.5-7B-Instruct | 7B | 29.57% ± 1.53% |
| 24 | ChatGLM | ChatGLM3-6B | 6B | 28.23% ± 0.58% |
| 25 | 🔒davinci | davinci-002 | ? | 21.25% ± 1.17% |
| 26 | Phi | Phi1.5 | 1.5B | 20.56% ± 0.09% |
| 27 | CodeGeeX | CodeGeeX2-6B | 6B | 19.88% ± 0.36% |
| 28 | CodeGen2 | CodeGen2-16B | 16B | 16.97% ± 1.15% |
| 29 | IEITYuan | Yuan2-51B-hf | 51B | 15.25% |
| 30 | CodeGen | CodeGen-16B-multi | 16B | 13.62% ± 1.18% |
| Human | | 10 Highest-Voted Answer Posts | | 65.18% |
| | | Highest-Voted Answer Post | | 56.28% |
| | | Officially-Accepted Answer Post | | 52.90% |

(b) InfiBench leaderboard by model type, where top five model within each type is shown.

| Type | Rank | Model Family / Model Name | Size | InfiBench Score |
|---|---|---|---|---|
| Proprietary Model | 1 | 🔒GPT-4/GPT-4-0613 | ? | 70.64% ± 0.82% |
| | 2 | 🔒GPT-4/GPT-4-turbo-1106 | ? | 68.42% ± 0.38% |
| | 3 | 🔒GPT-4/GPT-4o-2024-05-13 | ? | 66.19% |
| | 4 | 🔒Claude 3/Claude 3 Opus | ? | 63.89% |
| | 5 | 🔒Mistral/mistral-large | ? | 58.22% |
| Code Fine-tuned Model | 1 | DeepSeek Coder/deepSeek-coder-V2-instruct | 236B / 21B | 65.49% |
| | 2 | Mistral Open/Codestral-22b | 22B | 62.98% ± 0.56% |
| | 3 | DeepSeek Coder/deepseek-coder-33b-instruct | 33B | 62.96% |
| | 4 | Phind/Phind-CodeLlama-34B-v2 | 34B | 59.00% |
| | 5 | Phind/Phind-CodeLlama-34B-v1 | 34B | 58.47% |
| Code Base Model | 1 | Code Llama/CodeLlama-34b | 34B | 47.36% |
| | 2 | Code Llama/CodeLlama-34b-Python | 34B | 43.13% |
| | 3 | StarCoder2/15B | 15B | 42.52% ± 1.24% |
| | 4 | Code Llama/CodeLlama-13b | 13B | 41.66% ± 0.84% |
| | 5 | Code Llama/CodeLlama-13b-Python | 13B | 41.31% ± 0.90% |
| General Fine-tuned Model | 1 | DeepSeek LLM/deepseek-llm-67b-chat | 67B | 57.41% |
| | 2 | Mistral Open/mixtral-8x7B-Instruct | 46.7B / 12.9B | 55.55% |
| | 3 | Qwen/Qwen-72B-Chat | 72B | 52.97% |
| | 4 | 01.AI/Yi-34B-Chat | 34B | 49.58% |
| | 5 | Zephyr/Zephyr 7B beta | 7B | 46.31% ± 1.11% |
| General Base Model | 1 | Qwen/Qwen-72B | 72B | 55.34% |
| | 2 | Qwen/Qwen-14B | 14B | 43.69% ± 1.09% |
| | 3 | DeepSeek LLM/deepseek-llm-67b-base | 67B | 39.87% |
| | 4 | Llama 2/Llama2-70B | 70B | 37.69% |
| | 5 | Qwen/Qwen-7B | 7B | 31.69% ± 0.29% |

(c) InfiBench leaderboard by model size, where best model within the threshold is shown.

| Size Threshold | Model Family / Model Name | Size | InfiBench Score |
|---|---|---|---|
| ∞ | 🔒GPT-4/GPT-4-0613 | ? | 70.64% ± 0.82% |
| <100B | Mistral Open/Codestral-22b | 22B | 62.98% ± 0.56% |
| <20B | DeepSeek Coder/deepseek-coder-6.7b-instruct | 6.7B | 53.25% ± 0.40% |
| <5B | DeepSeek Coder/deepseek-coder-1.3b-instruct | 1.3B | 41.32% ± 1.12% |

MoE models are shown as horizontal segments with error ranges connecting the activated parameters during inference and total parameters, and strong proprietary models are shown as horizontal lines.

In both tables and the figure, we classify LLMs by general/code and base/finetuned. The general LLMs are claimed to have strong capabilities beyond code, e.g., in various natural language tasks, while the code LLMs are exclusively optimized for the code domain. The base LLMs only went through the pretraining phase, while the finetuned LLMs are claimed to have instruction-following capabilities or are finetuned on instruction or human preference datasets.

## 4 Analysis and Discussion

**The best model so far, GPT-4, is still far from perfect, and open-source models are competitive but still far from GPT-4.** GPT-4 achieves the highest score 70.64% (interestingly, achieved by GPT-4-0613 instead of the more recent GPT-4o), then Claude 3 Opus with a score 63.89%, and then Codestral-22b [1] with a score 62.98% and deepseek-coder-33b-instruct [15] with a score 62.96%. The result implies that: (1) Noting that the full score is 100%, even the powerful GPT-4 is still far from perfect, which is in contrast to its ≈90% HumanEval score. We inspect the score breakdown. For the two most frequent metric types, keywords matching and unit testing, GPT-4 achieves similar scores 66.61% and 76.00% respectively. For blank filling, the score is relatively lower at 58.08%. These scores imply that GPT-4 may still lack generic ability in answering diversified real-world questions related to code. When instructed to follow a given template to answer (blank filling), due to the more strict requirement and narrower solution space, its lack of capability is more pronounced. (2) There is still a visible gap between open-source models and GPT-4. The gap between the most powerful open-source model, Codestral-22b, and GPT-4 is roughly 8 points. On the other hand, noticing that GPT-3.5-turbo achieves 56.47%, the open-source model, Codestral-22b, is now reliably better than GPT-3.5-turbo with merely 22B parameters which is promising.

**Among open-source models, different models have various performances.** Figure 4 systematically visualizes the performance of different open-source models at diverse scales. Although there is a general tendency that larger models achieve higher scores, the scores among different models at a similar scale differ largely. For example, on scale 7B, the best-performing model is at around

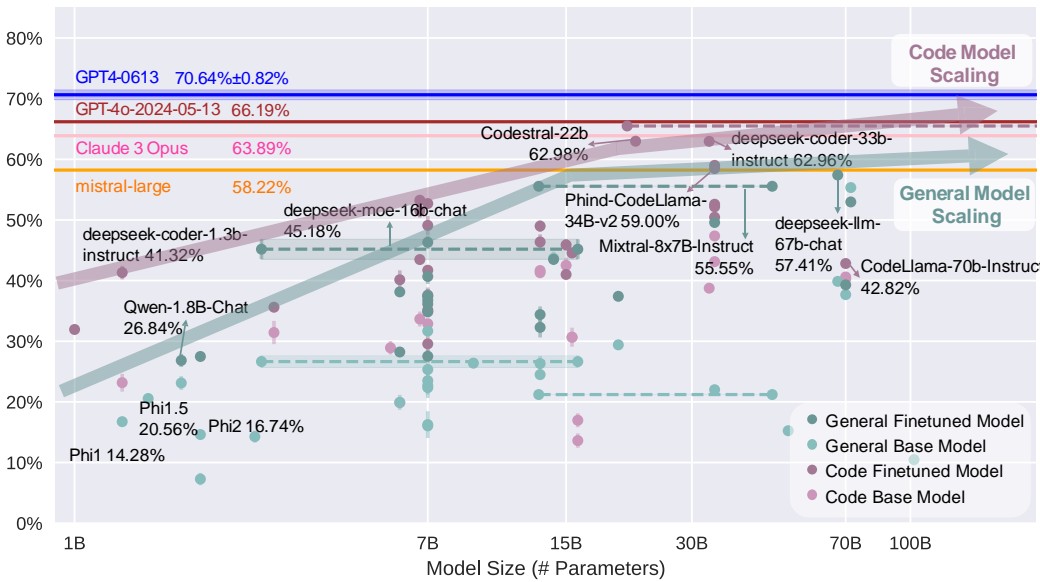

Figure 4: **Scatter plot for all evaluated LLMs on InfiBench**. $x$-axis is the model size in terms of number of parameters and $y$-axis is InfiBench score. Projected empirical scaling laws for both general and code models are drawn. Detail discussion in Section 4.

55%, pretty close to GPT-3.5, while the low-performing model stays at around 15%. Moreover, deepseek-coder-1.3b-instruct achieves 41.32% at 1.3B and surpasses a few models at scale 70B or 100B. Hence, though scaling matters, the training techniques and training data are equally important or even more, helping to reduce the required scale for achieving a certain score by more than $10\times$.

**Hard problems generalize their difficulties.** We rate the benchmark problem difficulty with five levels by how well GPT-4 and GPT-3.5-turbo answer them, as detailed in Appendix D. Example questions from each level are shown in Appendix I. We present the detail result table including the sub-score for each difficulty level in Appendix E. Interestingly, the trend is *highly consistent that sub-scores decrease along with the increase of problem level*. Specifically, hard problems for the most powerful model yet, GPT-4, are also generally hard for open-source models. These hard problems usually correspond to code generation with long and domain-specific context or challenging blank-filling questions since blank-filling is a specific task that rarely appears in training data before.

**Instruction finetuning is important for QA.** Among models of similar scales and the same family, we find that the best-performing ones almost always include an instruction-finetuning phase, such as deepseek-llm-67b-chat, deepseek-coder-33b-instruct, CodeLlama-34B-Instruct, and Qwen-18B-Chat. In contrast, the pretraining models, such as davinci-002 and phi models, usually perform poorly despite their strong performances in code generation benchmarks. Instruction-finetuning is also critical for other code domain tasks such as code generation. As shown in Appendix F.1 where we plot model scores in QA (measured by InfiBench) and code generation (measured by HumanEval pass@1), instruction-tuning generally improves both QA and code generation, but the improvement is usually more significantly on code generation but more moderately on QA. As a result, we suggest generalizing the instruction-finetuning data beyond simple coding problems to improve code LLMs. Indeed, our preliminary experiments show that, after fine-tuning with the decontaminated and sanitized Stack Overflow data, we improved InfiBench scores for Codellama-13b-Instruct from 46.37% to 60.74% and for mixtral-8x7B-Instruct from 55.55% to 62.61%.

**Some models may focus too much on code generation, especially the small ones.** As detailed in Appendix F.1, we observe that for large models (>30B) and top entries, InfiBench and HumanEval pass@1 scores coincide well. However, for smaller models, the score tendencies start to diverge, where some models are relatively stronger on InfiBench (Mixtral-8x7B-Instruct) and more are relatively stronger on HumanEval (Phi1, Phi2, gemma-7b, ...). This phenomenon implies that a few models may be optimized too heavily on code generation benchmarks while ignoring the performance in generic code scenarios as represented by InfiBench, which in turn highlights the significance of free-form QA benchmarks like InfiBench in detecting capability imbalance in code LLMs.

**Code LLama models have unique characteristics.** We evaluated all Code Llama models [39]. As shown in Table 5, we found finetuning on Python data improves on HumanEval but hurts InfiBench scores, while instruction finetuning usually improves InfiBench scores but may hurt HumanEval. As a side product, we found CodeLlama-70B may be overly safeguarded and denies answering some

Table 5: Evaluation on eight models from the Code Llama [39] family showcases intense Python finetuning may hurt free-form QA ability, despite achieving higher HumanEval scores.

| | Benchmark | Base | Python | Instruct |
|---|---|---|---|---|
| 7B | HumanEval | 33.5% | 38.4% (+4.9%) | 34.8% (+1.3%) |
| | InfiBench | 37.62%$_{\pm1.28\%}$ | 32.89%$_{\pm0.45\%}$ (−4.73%) | 35.15%$_{\pm1.28\%}$ (−2.47%) |
| 13B | HumanEval | 36.0% | 43.3% (+7.3%) | 42.7% (+6.7%) |
| | InfiBench | 41.66%$_{\pm0.84\%}$ | 41.31%$_{\pm0.90\%}$ (−0.35%) | 46.37%$_{\pm1.26\%}$ (+4.71%) |
| 34B | HumanEval | 48.8% | 53.7% (+4.9%) | 41.5% (−7.3%) |
| | InfiBench | 47.36% | 43.13% (−4.23%) | 50.45% (+3.09%) |
| 70B | HumanEval | 53.0% | 57.3% (+4.3%) | 67.8% (+14.8%) |
| | InfiBench | 40.60% | 40.29% (−0.31%) | 42.82% (+2.22%) |

safeguarded and denies answering some safe questions in InfiBench. More model-specific findings are presented in Appendix F.

**Code models and general models may exhibit different scaling laws, and open-source models scale well only within 40B yet.** In Figure 4, we use the top-performing code and general models at each scale respectively to regress and extrapolate model performance at larger scales. As shown, code models tend to have higher capabilities compared to general models of the same scale, though the gap shrinks for larger models. Hence, when the compute budget is heavily limited, training exclusively in the code domain could be more efficient for building strong code LLMs.

In Figure 4, both predicting curves are split into two segments, steep in the first segment and much flat in the second. Following the first segment, open-source models catch up with GPT-4 at around 50B scale. However, following the second segment, they may need to be at >300B scale to catch up. The finding contradicts the common scaling law [20, 34, 7] where a strong linear relationship between model scale and capability exists. The contradiction implies that very large open-source models (>40B) may fail to achieve the expected performance at their scales, or there is some non-trivial barrier when scaling the model beyond 40B, or the scaling law may change at such a large scale. We leave further investigation as the future work. Notably, after the release of InfiBench, Deepseek-coder-v2 [49] was released as the largest code LLM to our knowledge in an MoE architecture with 236B total and 21B active parameters. On InfiBench, Deepseek-coder-v2 achieves 65.49%, setting the new baseline for open-source LLMs but still being inferior to GPT-4. More importantly, the score is within the predicted range of our empirical scaling law.

We defer dataset card and data accessibility details, discussion on limitations and societal impact, full leaderboard, additional findings, ablation studies, and data examples in appendices.

## 5   Related Work

Large language models [44, 11, 8] are transforming people's lives. In the coding domain, LLMs [9, 24] are shown to be capable of completing a wide range of tasks such as code generation, debugging, and question-answering. Recently, code LLMs are booming. New models, including both proprietary [14, 36] and open-source ones [4, 35, 42, 43, 22, 30, 39, 49], emerge almost every month.

Benchmarks for code LLMs are developing, though at a relatively slower pace. Common benchmarks, e.g., APPS [16], MBPP [3], and HumanEval [9], focus on code generation and unit-test-based evaluation. Some efforts augment these benchmarks by language translation (e.g., Multilingual HumanEval [2], HumanEval-X [48]), test augmentation (e.g., HumanEval+ [26]), task generalization (e.g., HumanEvalPack [33]), and human rewriting (e.g., LBPP [32]). To systematically evaluate real-world problem solving, recently, SWE-bench [19], its filtered version SWE-bench Verified [38], and RepoBench [27] are proposed but they still primarily focus on code generation. Some general-purpose benchmarks, e.g., Arena-Hard [23], contain code-related questions, but rely on LLM to judge and do not provide domain-specific scores. CodeXGLUE [29] considers multiple coding capabilities beyond code generation, but replies on existing data sources. In contrast to these benchmarks, InfiBench benchmark is built for evaluating free-form question-answering ability in the code domain beyond code generation in an automated and model-independent way.

## 6   Conclusion

We proposed InfiBench, a systematic benchmark for evaluating the question-answering ability of code LLMs in real-world scenarios, to facilitate development and scientific evaluation of LLMs. InfiBench comprises 234 high-quality questions from Stack Overflow and supports automatic model-free evaluation. A comprehensive evaluation of over 100 code LLMs reveals several findings and takeaways. The benchmark is publicly available and continuously expanding.

## Acknowledgement

We thank ByteDance Inc. for the support on computing resources, anonymous reviewers for their constructive feedback, and Kaixin Li (National University of Singapore) for contributing the Docker image after the initial release of InfiBench. This work was partially supported by National Natural Science Foundation of China under Grant No. 62161146003, and the Tencent Foundation/XPLORER PRIZE. Tao Xie is also affiliated with the School of Computer Science, Peking University, China. The corresponding authors are Linyi Li and Tao Xie.

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

# Appendices

In appendices, we present dataset card and data accessibility details in Appendix A, discussion on limitations and societal impact in Appendix B, agreement statistics between InfiBench/GPT-4o and human in Appendix C, question grouping by difficulty in Appendix D, full leaderboard in Appendix E, additional findings in Appendix F, study of sampling hyperparameters in Appendix G, prompts in Appendix H, and benchmark data examples in Appendix I.

## A    Dataset Card and Accessibility Details

> ### Dataset Card
>
> - **Name**: InfiBench
> - **Description**: Evaluation Dataset for the Question-Answering Capabilities of Code Large Language Models
> - **URL**: `https://infi-coder.github.io/infibench` (all resources) / `https://huggingface.co/datasets/llylly001/InfiBench` (data part)
> - **Version**: 2.1
> - **License**: Creative Commons Attribution Share Alike 4.0
> - **Citation**:
>
>   ```
>   @misc{infibench,
>       title={InfiBench: Evaluating the Question-Answering Capabilities
>       of Code Large Language Models},
>       howpublished = "\url{https://infi-coder.github.io/infibench}",
>       author={InfiBench},
>       year={2024}
>   }
>   ```
>
> - **DOI**: `doi:10.57967/hf/2474`
> - **Responsible AI — Data Collection**:
>   Data source is downloaded from the publicly available StackExchange archive (`https://archive.org/download/stackexchange`, `https://ia904700.us.archive.org/view_archive.php?archive=/6/items/stackexchange/stackoverflow.com-Posts.7z`). Especially, we use the preprocessed version from `https://huggingface.co/datasets/mikex86/stackoverflow-posts` where all posts are formatted in Markdown text.
>
>   We choose to keep only the questions with at least three positively voted answers and an officially accepted answer, which turn out to be 1,090,238 questions. For these one million questions, we choose to keep frequently viewed and relatively new questions.
>
>   Utilizing the mandatory question tags of these questions, we then manually construct a tag tree that covers the 200 most frequent tags, enabling us to identify the top programming languages and areas for 14,330105 out of these 17,402 questions. We exclude 6 programming languages that either describe data or are domain-specific: JSON, regex, Markdown, YAML, CSV, and SQL. As a result, we compile 13,854 questions that serve as the initial seed set.
>
>   We randomly sample from the initial seed set. Then we recruited five domain experts inside our company to create the benchmark from the sampled initial seed set, each in charge of one area. The annotation process is composed of three steps: (1) Question Selection and Type Annotation; (2) Prompt Paraphrasing. (3) Correctness Criterion Annotation.
> - **Responsible AI — Data Biases**:
>   The data essentially serves as an evaluation benchmark. We foresee data biases in the following aspects:
>
>   (1) Non-standard evaluation. Alongside the data is a comprehensive benchmark of existing code LLMs. The benchmark scores are evaluated under a specific set of hyperparameters

(e.g, temperature 0.2, top probability 0.9, best@10 at question level). Data usage under different evaluation conditions may result in misleading comparison results and conclusions.

(2) Usage misinterpretation. The benchmark focuses on evaluating the response correctness of code LLMs for a set of real-world developers' questions. Our evaluation standard does not specifically take other aspects (naturalness, conciseness, fairness, politeness, etc) into consideration. Hence, this is risk of overinterpreting the evaluation results. When evaluating a code LLM, we recommend combining this benchmark score with other evaluations to be a more comprehensive evaluation.

(3) Potential data contamination. Though we have made our efforts to reduce the impact of data contamination, future code LLMs may train or fine-tune on this benchmark dataset to improve the score on InfiBench. This could be challenging to prevent as a cost of being fully public. On the other hand, as responsible LLM developers, we hope future practitioners would report how they use the benchmark data if beyond the original scope (for evaluation use).

- **Responsible AI — Personal Sensitive Information**: During the data construction process, our domain experts paraphrased the question prompts to remove personal and sensitive information (PII) and a cross validation stage was introduced to further ensure the PII removal.

**Croissant Dataset Description:** `https://huggingface.co/datasets/llylly001/InfiBench/blob/main/croissant-infibench.json`. Note that the Croissant format is mainly designed for machine learning dataset description. However, InfiBench is more than a dataset; it is an evaluation benchmark including response evaluation standards, tools, and an accompanying leaderboard. Hence, the Croissant script records only the CSV file and covers question prompts and evaluation standards; whereas the open-source evaluation tool and leaderboard are not recorded which can be separately downloaded from `https://infi-coder.github.io/infibench`.

**Data Accessibility.** As briefly mentioned in the main text, all materials are made publicly available and accessible at the website: `https://infi-coder.github.io/infibench` without personal request. The materials include three parts: (1) Benchmark questions and evaluation metrics — this part is additionally uploaded to Hugging Face (URL and DOI are in the above dataset card). (2) Automatic evaluation tool — this part is uploaded and maintained in a dedicated GitHub repo `https://github.com/infi-coder/infibench-evaluator`. In addition, we uploaded our extension of `bigcode-evaluation-harness` [5], namely `infibench-evaluation-harness` to a dedicated GitHub repo `https://github.com/infi-coder/infibench-evaluation-harness`. The extension includes the inference code on InfiBench for all evaluated LLMs. (3) Evaluation raw data and leaderboard — the leaderboard is displayed on the website `https://infi-coder.github.io/infibench` and the raw model responses are stored in the website repo `https://github.com/infi-coder/infibench`. All materials are under the Creative Commons Attribution Share Alike 4.0 license. In the above dataset card and Appendix B, we anticipate potential inappropriate usage of the benchmark and we encourage the practitioners to document their usage of the benchmark if beyond model evaluation. In the future, we will continue the maintenance and expansion of the benchmark. Furthermore, we are developing an adaptor for automatic evaluation on Hugging Face so that InfiBench can be integrated into the Hugging Face Open LLM Leaderboard [4] to further ease the evaluation burden.

# B  Limitations, Societal Impacts, and Future Work

In this appendix, we expand our discussion of limitations, potential societal impacts, and future work.

**Evaluation Metric.** In InfiBench, the expert-annotated evaluation metric is designed to mainly focus on response correctness, more specifically, whether the response contains key information that solves the given question. Concretely, the metric may evaluate whether the response passes a given set of unit tests, whether it suggests the right API or concept, whether it follows the instruction to provide relevant information, etc. Hence, the score comes with two limitations: (1) The score is subjective since the metric is annotated by human experts without an explicit and universal

Table 6: Confusion matrices between InfiBench/GPT-4o and human. Details in Appendix C. Bolded cells correspond to when both methods have clear preferences on one response.

(a) Between InfiBench and human.

| | | Human | | | | |
|---|---|---|---|---|---|---|
| | | $A > B$ | $B > A$ | $A \approx B \uparrow$ | $A \approx B \downarrow$ | Tot. |
| InfiBench | $A > B$ | **23** | **3** | 9 | 4 | 39 |
| | $B > A$ | **4** | **17** | 12 | 2 | 35 |
| | $A \approx B \uparrow$ | 0 | 0 | 10 | 0 | 10 |
| | $A \approx B \downarrow$ | 4 | 3 | 3 | 6 | 16 |
| | Tot. | 31 | 23 | 34 | 12 | 100 |

(b) Betwen GPT-4o and human.

| | | Human | | | | |
|---|---|---|---|---|---|---|
| | | $A > B$ | $B > A$ | $A \approx B \uparrow$ | $A \approx B \downarrow$ | Tot. |
| GPT-4o | $A > B$ | **23** | **7** | 8 | 6 | 44 |
| | $B > A$ | **3** | **12** | 9 | 3 | 27 |
| | $A \approx B \uparrow$ | 5 | 4 | 15 | 3 | 27 |
| | $A \approx B \downarrow$ | 0 | 0 | 2 | 0 | 2 |
| | Tot. | 31 | 23 | 34 | 12 | 100 |

standard. Note that we did not aim to provide an objective metric since the developers' views of response correctness intrinsically vary and diverge for these diverse questions. On the other hand, we introduce a cross-validation and calibration stage to improve the metric representativeness of most developers' standards. We leave it as a future work to further quantitatively measure and improve the metric representativeness. (2) The score focuses mainly on correctness. Several other aspects define a model's usability, such as language naturalness (including conciseness, politeness, etc), trustworthiness (refusal of risky questions, fairness, unbiasedness, privacy, etc), and system-level metrics (latency, throughput, parallelism-friendliness, etc). Model evaluators and practitioners may keep in mind that InfiBench score is not a comprehensive usability measurement of code LLMs, and we strongly encourage them to combine InfiBench score with benchmarks on these other aspects (c.f. [6, 45]) to comprehensively evaluate LLMs.

**Data Contamination.** The limitations and mitigations on data contamination are discussed in Section 2.5. In addition, as a side effect of open source, future code LLMs may leverage the benchmark data to deliberately introduce data contamination to achieve a high score in InfiBench. To partly detect such data contamination, our evaluation of using the original stack Overflow answers might be a proxy. According to Table 4(a), even gold extraction from human answers cannot saturate the benchmark while strong LLMs like GPT-4 surpassed human answers. Hence, if a future model achieves scores close to human answers (between 50% and 65%) but cannot further improve beyond human along with scaling, data contamination may potentially happen. Detecting data contamination is itself a research topic where research on member inference attacks [40, 31] is involved. We did not integrate a detection module in the current release of InfiBench but we are planning to inspect this topic in the future.

**Labelling Cost.** InfiBench construction involves human labelling cost, where domain experts paraphrase the source question post and label the evaluation metric. Such a cost prevents the InfiBench from scaling up in terms of size, and the questions for less popular programming languages, such as Rust and Ruby, are relatively few. In an attempt to mitigate this limitation, we explored a few alternative evaluation metrics, such as dialogue similarity with officially accepted answers. However, these alternatives either require a language model which may induce bias and heavy computing cost, or deviate away from domain experts' correctness judgment. We leave the exploration of more scalable metrics and annotation procedures as future work and make the benchmark fully open source so community involvement may boost the expansion.

## C  Agreement Statistics between InfiBench/GPT-4o Evaluation and Human

In Section 2.4, we evaluated the alignment between InfiBench/GPT-4o evaluation and human evaluation by generating 100 response pairs for InfiBench questions and let InfiBench, GPT-4o, and human annotators to grade into four outcomes.

Table 6 shows the confusion matrices between InfiBench/GPT-4o and human, where each cell corresponds to the frequency of each combination of outcomes among 100 pairs. The implication of each outcome is introduced in Section 2.4.

Learned from Table 6, if we only count the cases where both human and InfiBench have clear preferences, their agreement rate is $\frac{40}{47} = 85.1\%$; if we only count the cases where both human and GPT-4o have clear preferences, their agreement rate is $\frac{35}{45} = 77.8\%$. Hence, the InfiBench evaluation aligns with human experts better than the GPT-4o evaluation (with >80% confidence). Furthermore,

we observe that GPT-4o has a stronger opinion and tends to choose one response more often, so it falls short when $A$ and $B$ are both bad responses, labelling none of them as "both bad". We also observe that InfiBench evaluation could be too strict due to pattern matching and fixed post-processing leading to over-differentiation—when a human believes $A$ and $B$ are both good responses, with only a 29.4% chance InfiBench labels them as "both good".

## D    Difficulty Grouping

We systematically evaluated GPT-4 and GPT-3.5-turbo on the benchmark following the evaluation protocol in Section 3, based on which we classify the benchmark questions into five disjoint difficulty groups.

- Level 1 (93 questions, 39.7%): GPT-3.5-turbo can achieve a mean score $\geq 0.5$.
- Level 2 (55 questions, 23.5%): Among the rest questions, those where GPT-4's mean score $\geq 0.5$.
- Level 3 (44 questions, 18.8%): Among the rest questions, those where GPT-4 with sampling temperature 1.0 can achieve a maximum score $\geq 0.5$ among 10 trials.
- Level 4 (18 questions, 7.7%): Among the rest questions, those GPT-4 with sampling temperature 0.2 can achieve a positive score among 100 trials.
- Level 5 (24 questions, 10.3%): The remaining questions, i.e., GPT-4 cannot get score among 100 trials.

Appendix E shows each code LLM's score in each difficulty group. The mean scores strictly decrease for higher difficulty levels, highlighting that the question difficulty is in general consistent across different code LLMs and our group assignment is reasonable. We hope that the grouping can help better reveal the strengths and weaknesses of a code LLM for different questions.

Question examples by difficulty groups are in Appendix I.

## E    Evaluation Details and Full Benchmark Results

**Evaluation Details of Code LLMs.**    For proprietary model evaluation, we did not specify the max tokens to generate and found out that the longest response generated by GPT-4 has 662 tokens with Code Llama tokenizer.

For open-source model evaluation, for models with over 30B parameters, due to the GPU memory limit and efficiency concerns, we impose the longest context constraint of 4,096 tokens and experiment just once. Since there is only one question whose GPT-4 context (prompt + GPT-4 response) can exceed 4,096 tokens, we think this context constraint has little effect, reducing the score by 0.37% at most. For models within 30B parameters, since GPT-4 response has at most 662 tokens, we set the max number of tokens to generate to be $\min\{1024, \text{context length - prompt length}\}$, providing some wiggle room. Meanwhile, we repeat the evaluation three times for models within 30B parameters.

**Evaluation Details of Original Stack Overflow Answers.**    As listed in Table 4(a) and Table 7, besides evaluating LLM responses, we evaluated the score of human-written original Stack Overflow answers since the question prompts are paraphrased from Stack Overflow. We consider three settings: (1) evaluating the officially-accepted answer post (note that we select only the Stack Overflow questions with an officially-accepted answer into the benchmark); (2) evaluating the highest-voted answer post (note that any registered user can equally vote for or against an answer); and (3) evaluating the highest-voted answer posts up to 10 and recording the highest score achieved by any post. For the last setting, we chose the number 10 because the main evaluation metric of model response is best@10. Moreover, we observe that all officially accepted answers for InfiBench questions are among the top 10 highest-voted answer posts. Note that there is no randomness of scores from Stack Overflow answers, so we do not repeat the evaluation nor report the standard deviation.

As expected, the last setting achieves the highest score 65.18% among the three settings. Due to its consistency with models' evaluation metric best@10, we deem this score most comparable with scores from LLMs. Interestingly, when considering only one answer post, the second setting, selecting the highest-voted answer, is better than the first setting, selecting the officially accepted answer.

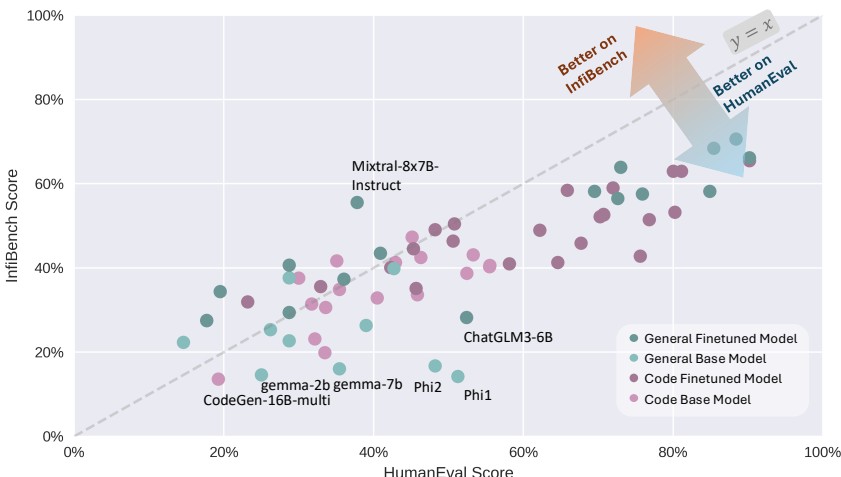

Figure 5: InfiBench and HumanEval scores as a scatter plot for LLMs. $r = 0.8058$. Discussion in Appendix F.1.

**Full Benchmark Results.** We present the full leaderboard in Table 7 (by descending order of InfiBench scores) and Table 8 (by alphabetical order of model family names). These tables are expanded from the aggregated Table 4. In these tables, we show model properties including size and context length. We also present HumanEval [3] scores since HumanEval is one of the most widely used benchmarks for evaluating code LLMs (further discussion in Appendix F). Furthermore, we represent the score breakdown by difficulty levels, problem types, and evaluation metric types. The proportion of each difficulty level can be found in Appendix D, and the proportion of each problem type and evaluation metric type is shown in Table 3(a,b). InfiBench score can be computed by the weighted sum of breakdown subscores by proportions. We present the score of human-written original Stack Overflow answers in the last three rows.

In tables, the mean scores are computed from scores of all 106 code LLMs. We observe that the mean overall score, 37.82%, is still much inferior to human answers (which achieves over 50% even with just one attempt). The model performance is monotonically decreasing for higher difficulty levels; relatively equivalent across different problem types; and weaker under blank-filling and dialogue-similarity metrics than keyword-matching and unit-testing metrics.

# F Additional Findings and Discussion

In this appendix, we present additional findings and discussion that are omitted from Section 3.

## F.1 Correlations between InfiBench and HumanEval Scores

We study the correlation between InfiBench and HumanEval pass@1 scores for different LLMs. In Figure 5, we plot LLMs with both InfiBench and HumanEval scores, in total 66 LLMs, in Table 7 as a scatter plot. The figure shows that scores on the two benchmarks are generally positively correlated, with a Pearson correlation coefficient $r = 0.8058$. If conducting a linear regression, we would observe that different model types (i.e., general/code model, base/finetuned model) share almost the same linear relationship, indicating that both benchmarks can reflect the model capability in general. Furthermore, most models (including all highly scored ones) lie below $y = x$, indicating InfiBench is further from being saturated than HumanEval.

However, a few outlier models exist in Figure 5. Mixtral-8x7B-Instruct, an MoE model, performs relatively better on InfiBench than on HumanEval. Some other models, e.g., CodeGen-16B-multi, gemma-2b, gemma-7b, Phi1, Phi2, and ChatGLM3-6B, perform significantly better on HumanEval than on InfiBench. These models are relatively small or old-dated. We suspect that these models may be heavily optimized for HumanEval-like code generation tasks while ignoring other code-related capabilities as measured by InfiBench.

Table 7: **Full leaderboard of all benchmarked LLMs ranked by InfiBench scores**. Evaluation protocol in Section 3 and details explained in Appendix E. Icon "🔒" stands for proprietary models otherwise open-source. As a reference, HumanEval scores digested from [26] and each model's report are shown. Bar colors stand for General Base , General Finetuned , Code Base , and Code Finetuned models respectively. Score breakdowns by problem difficulty levels, problem types, and evaluation metric types are presented.

| Rank | Model Family | Model Name | Size (# Param.) | Context Length | InfiBench Score | HumanEval | Difficulty Levels | | | | | Problem Type | | | | Evaluation Metric Type | | | |
|---|---|---|---|---|---|---|---|---|---|---|---|---|---|---|---|---|---|---|---|
| | | | | | | | Level 1 | Level 2 | Level 3 | Level 4 | Level 5 | Code Completion | Code Debugging | Knowledge QA | Config & Env Debugging | Keyword Matching | Unit Testing | Blank Filling | Dialogue Similarity |
| 1 | 🔒GPT-4 | GPT-4-0613 | ? | 8192 | 70.64% ± 0.82% | 88.4 | 92.31% | 92.48% | 51.90% | 31.91% | 0.00% | 75.23% | 69.74% | 68.55% | 66.63% | 66.61% | 76.00% | 58.08% | 84.27% |
| 2 | 🔒GPT-4 | GPT-4-turbo-1106 | ? | 8192 | 68.42% ± 0.38% | 85.4 | 89.90% | 78.57% | 54.16% | 30.93% | 16.20% | 74.82% | 65.36% | 67.47% | 62.98% | 64.98% | 76.40% | 53.91% | 52.85% |
| 3 | 🔒GPT-4 | GPT-4o-2024-05-13 | ? | 8192 | 66.19% | 90.2 | 91.29% | 78.46% | 46.43% | 28.05% | 5.21% | 75.00% | 63.32% | 65.65% | 61.70% | 61.99% | 76.40% | 53.91% | 70.73% |
| 4 | DeepSeek Coder | deepSeek-coder-V2-instruct | 236B / 21B | 128000 | 65.49% | 90.2 | 88.77% | 76.97% | 50.58% | 17.31% | 12.50% | 74.77% | 63.89% | 59.57% | 59.30% | 58.91% | 76.00% | 55.77% | 37.34% |
| 5 | 🔒Claude 3 | Claude 3 Opus | ? | 200000 | 63.89% | 73 | 84.36% | 78.95% | 39.98% | 31.76% | 18.06% | 65.18% | 62.94% | 65.86% | 60.49% | 60.07% | 61.80% | 59.36% | 44.91% |
| 6 | Mistral Open | Codestral-22b | 22B | 32768 | 62.98% ± 0.56% | 81.1 | 88.64% | 69.90% | 49.97% | 17.11% | 5.90% | 68.75% | 63.65% | 61.07% | 54.28% | 57.72% | 73.33% | 45.92% | 57.08% |
| 7 | DeepSeek Coder | deepseek-coder-33b-instruct | 33B | 16384 | 62.96% | 80.02 | 87.58% | 72.02% | 44.12% | 15.83% | 16.67% | 71.26% | 57.14% | 63.14% | 56.81% | 59.01% | 77.00% | 30.00% | 36.09% |
| 8 | Phind | Phind-CodeLlama-34B-v2 | 34B | 4096 | 59.00% | 71.95 | 83.67% | 55.57% | 53.12% | 15.09% | 14.93% | 58.24% | 58.30% | 63.60% | 55.33% | 59.63% | 58.40% | 35.26% | 24.19% |
| 9 | Phind | Phind-CodeLlama-34B-v1 | 34B | 4096 | 58.47% | 65.85 | 81.38% | 63.85% | 47.05% | 22.63% | 5.21% | 66.13% | 56.94% | 56.79% | 49.48% | 55.71% | 66.00% | 38.78% | 35.39% |
| 10 | 🔒Mistral | mistral-large | ? | 32768 | 58.22% | 69.5 | 81.76% | 66.59% | 41.66% | 23.62% | 4.17% | 66.69% | 50.10% | 60.21% | 52.89% | 53.17% | 67.00% | 45.64% | 42.66% |
| 11 | 🔒Claude 3 | Claude 3 Sonnet | ? | 200000 | 58.20% | 84.9 | 80.13% | 65.55% | 42.48% | 18.06% | 15.28% | 62.61% | 52.34% | 63.61% | 52.12% | 54.22% | 66.00% | 46.35% | 25.62% |
| 12 | 🔒Claude 3 | Claude 3 Haiku | ? | 200000 | 57.57% | 75.9 | 79.86% | 66.06% | 40.23% | 21.76% | 10.42% | 61.71% | 48.68% | 62.85% | 56.71% | 55.78% | 58.40% | 44.62% | 36.40% |
| 13 | DeepSeek LLM | deepseek-llm-67b-chat | 67B | 4096 | 57.41% | / | 82.96% | 63.03% | 39.09% | 22.60% | 5.21% | 61.42% | 52.73% | 58.72% | 55.63% | 53.14% | 63.00% | 51.41% | 36.68% |
| 14 | 🔒GPT-3.5 | GPT-3.5-turbo-0613 | ? | 4096 | 56.47% ± 1.34% | 72.6 | 93.08% | 49.77% | 31.36% | 14.30% | 7.64% | 64.91% | 48.50% | 59.47% | 49.64% | 51.28% | 70.07% | 40.90% | 40.13% |
| 15 | 🔒Mistral | mistral-small | ? | 32768 | 55.62% ± 0.46% | / | 82.19% | 55.98% | 35.72% | 22.58% | 10.07% | 63.56% | 44.12% | 64.13% | 47.75% | 50.56% | 68.00% | 39.08% | 53.32% |
| 16 | Mistral Open | mixtral-8x7B-Instruct | 46.7B / 12.9B | 32768 | 55.55% | 37.8 | 82.19% | 56.72% | 31.53% | 24.00% | 17.36% | 54.01% | 51.57% | 63.69% | 53.59% | 56.14% | 50.40% | 35.58% | 61.75% |
| 17 | Qwen | Qwen-72B | 72B | 32768 | 55.34% | / | 81.98% | 57.40% | 41.61% | 13.24% | 4.17% | 61.06% | 53.16% | 58.79% | 44.03% | 50.43% | 64.00% | 45.96% | 36.41% |
| 18 | DeepSeek Coder | deepseek-coder-6.7b-instruct | 6.7B | 16384 | 53.25% ± 0.40% | 80.22 | 77.88% | 56.30% | 35.18% | 18.89% | 9.72% | 65.95% | 46.44% | 52.46% | 42.12% | 48.24% | 70.40% | 26.90% | 23.48% |
| 19 | Qwen | Qwen-72B-Chat | 72B | 32768 | 52.97% | / | 82.44% | 47.00% | 36.09% | 18.34% | 9.38% | 58.67% | 43.31% | 60.12% | 44.31% | 49.26% | 59.00% | 43.08% | 33.95% |
| 20 | Magicoder | Magicoder-S-CL-7B | 7B | 16384 | 52.71% ± 0.72% | 70.7 | 77.97% | 50.42% | 40.20% | 13.45% | 12.50% | 51.39% | 51.98% | 56.97% | 50.58% | 53.28% | 56.67% | 21.41% | 26.97% |
| 21 | WizardLM | WizardCoder-Python-34B-V1.0 | 34B | 16384 | 52.59% | 70.73 | 78.51% | 52.50% | 34.25% | 20.05% | 10.42% | 60.32% | 46.39% | 55.86% | 44.01% | 48.73% | 64.00% | 37.56% | 24.72% |
| 22 | Phind | Phind-CodeLlama-34B-Python-v1 | 34B | 4096 | 52.17% | 70.22 | 80.54% | 48.44% | 42.58% | 8.57% | 1.04% | 54.41% | 52.34% | 57.11% | 41.47% | 51.04% | 57.80% | 27.18% | 39.76% |
| 23 | Magicoder | Magicoder-DS-6.7B | 6.7B | 16384 | 51.46% ± 1.09% | 76.8 | 78.93% | 51.02% | 28.91% | 25.93% | 6.48% | 62.54% | 46.45% | 55.74% | 33.84% | 45.64% | 69.13% | 31.45% | 27.86% |
| 24 | Code Llama | CodeLlama-34b-Instruct | 34B | 16384 | 50.45% | 50.79 | 72.60% | 55.07% | 33.16% | 18.43% | 9.72% | 51.71% | 48.37% | 61.36% | 37.04% | 48.14% | 51.20% | 47.76% | 28.55% |
| 25 | 01.AI | Yi-34B-Chat | 34B | 4096 | 49.58% | / | 76.81% | 47.15% | 29.32% | 26.39% | 4.17% | 44.10% | 44.75% | 62.29% | 49.84% | 53.15% | 45.30% | 36.15% | 33.07% |
| 26 | WizardLM | WizardCoder-Python-7B-V1.0 | 7B | 16384 | 49.10% ± 1.59% | 48.2 | 76.42% | 48.08% | 29.09% | 12.50% | 9.72% | 58.60% | 41.63% | 50.67% | 41.49% | 46.38% | 59.40% | 25.30% | 23.00% |
| 27 | WizardLM | WizardCoder-Python-13B-V1.0 | 13B | 16384 | 48.99% ± 0.92% | 62.19 | 76.21% | 46.76% | 34.19% | 16.17% | 0.35% | 52.69% | 44.88% | 57.33% | 41.32% | 48.71% | 53.73% | 20.45% | 29.61% |
| 28 | Code Llama | CodeLlama-34b | 34B | 16384 | 47.36% | 45.11 | 72.07% | 43.34% | 29.32% | 21.20% | 13.54% | 53.74% | 50.09% | 51.52% | 26.99% | 43.18% | 57.33% | 37.37% | 24.85% |
| 29 | Code Llama | CodeLlama-13b-Instruct | 13B | 16384 | 46.37% ± 1.26% | 50.6 | 69.07% | 45.99% | 34.37% | 11.42% | 7.52% | 48.65% | 45.18% | 49.67% | 39.83% | 47.71% | 50.47% | 20.90% | 12.45% |
| 30 | Zephyr | Zephyr 7B beta | 7B | 32768 | 46.31% ± 1.11% | / | 68.41% | 49.99% | 31.11% | 14.99% | 3.59% | 44.26% | 44.86% | 54.89% | 40.85% | 49.28% | 35.07% | 27.91% | 27.66% |
| 31 | StarCoder2 | 15B-Instruct | 15B | 16384 | 45.89% ± 0.95% | 67.7 | 70.37% | 50.21% | 24.15% | 11.44% | 6.83% | 56.02% | 38.52% | 46.30% | 38.56% | 40.55% | 60.27% | 25.21% | 45.01% |
| 32 | DeepSeek MoE | deepseek-moe-16b-chat | 16B / 2.8B | 16384 | 45.18% ± 1.65% | / | 68.15% | 46.72% | 27.55% | 10.71% | 11.23% | 47.19% | 46.54% | 45.58% | 39.09% | 45.71% | 44.73% | 25.85% | 20.70% |
| 33 | OctoPack | OctoCoder | 15.5B | 8192 | 44.55% ± 0.79% | 45.3 | 68.19% | 41.61% | 29.39% | 12.96% | 11.11% | 46.56% | 37.62% | 53.57% | 39.56% | 44.18% | 47.07% | 20.09% | 39.20% |
| 34 | Qwen | Qwen-14B | 14B | 8192 | 43.69% ± 1.09% | / | 67.61% | 47.64% | 21.87% | 9.63% | 7.52% | 44.59% | 42.15% | 47.09% | 39.99% | 41.61% | 44.40% | 34.19% | 28.21% |
| 35 | Qwen | Qwen-14B-Chat | 14B | 8192 | 43.49% ± 0.63% | 40.9 | 68.91% | 36.25% | 27.73% | 10.28% | 15.90% | 45.39% | 42.12% | 46.33% | 38.48% | 41.87% | 42.73% | 36.18% | 34.79% |
| 36 | Magicoder | Magicoder-DS-6.7B | 6.7B | 16384 | 43.47% ± 0.21% | / | 67.94% | 48.33% | 23.11% | 13.64% | 0.69% | 52.73% | 40.42% | 48.14% | 25.61% | 38.37% | 56.73% | 29.81% | 38.07% |
| 37 | Code Llama | CodeLlama-34b-Python | 34B | 16384 | 43.13% | 53.29 | 66.20% | 40.76% | 36.06% | 6.94% | 0.00% | 50.14% | 40.48% | 43.64% | 34.13% | 40.40% | 51.00% | 27.63% | 16.67% |
| 38 | Code Llama | CodeLlama-70b-Instruct | 70B | 4096 | 42.82% | 75.6 | 59.08% | 44.14% | 38.48% | 12.22% | 7.64% | 38.20% | 44.99% | 46.87% | 42.38% | 48.34% | 32.00% | 16.09% | 5.62% |
| 39 | StarCoder2 | 15B | 15B | 16384 | 42.52% ± 1.24% | 46.3 | 64.99% | 41.67% | 29.02% | 13.73% | 3.70% | 47.00% | 37.31% | 46.76% | 36.87% | 43.86% | 42.20% | 18.44% | 0.00% |
| 40 | Magicoder | Magicoder-CL-7B | 7B | 16384 | 41.71% ± 0.76% | / | 70.38% | 36.48% | 23.06% | 10.33% | 0.35% | 49.26% | 35.11% | 45.41% | 33.47% | 37.85% | 52.27% | 19.91% | 39.21% |
| 41 | Code Llama | CodeLlama-13b | 13B | 16384 | 41.66% ± 0.84% | 35.07 | 62.77% | 40.40% | 31.11% | 7.97% | 7.41% | 38.17% | 44.56% | 43.00% | 41.72% | 45.44% | 34.80% | 14.79% | 2.47% |
| 42 | DeepSeek Coder | deepseek-coder-1.3b-instruct | 1.3B | 16384 | 41.32% ± 1.12% | 64.6 | 65.48% | 41.42% | 25.48% | 6.30% | 2.78% | 41.80% | 42.56% | 42.88% | 36.38% | 41.00% | 45.20% | 16.52% | 24.32% |
| 43 | Code Llama | CodeLlama-13b-Python | 13B | 16384 | 41.31% ± 0.90% | 42.89 | 62.93% | 40.80% | 28.61% | 10.37% | 5.21% | 49.95% | 44.60% | 36.68% | 27.22% | 40.58% | 51.07% | 11.92% | 13.64% |
| 44 | WizardLM | WizardCoder-15B-V1.0 | 15B | 2048 | 41.01% ± 0.22% | 58.12 | 66.19% | 40.34% | 21.72% | 12.42% | 1.74% | 44.80% | 34.54% | 47.68% | 35.29% | 38.43% | 47.60% | 22.31% | 35.01% |
| 45 | 🔒Mistral | mistral-medium | ? | 32768 | 40.95% ± 0.41% | / | 72.59% | 30.34% | 19.14% | 8.15% | 7.29% | 41.49% | 34.39% | 49.19% | 39.09% | 38.24% | 42.67% | 33.85% | 18.26% |
| 46 | gemma | gemma-7b-it | 7B | 8192 | 40.68% ± 1.23% | 28.7 | 60.94% | 42.94% | 28.86% | 5.75% | 4.86% | 42.60% | 36.37% | 47.75% | 34.52% | 40.68% | 41.40% | 19.04% | 30.44% |
| 47 | Code Llama | CodeLlama-70b | 70B | 4096 | 40.60% | 55.5 | 60.59% | 37.42% | 35.68% | 7.59% | 4.17% | 47.18% | 39.10% | 39.09% | 33.21% | 40.54% | 45.00% | 19.23% | 8.56% |
| 48 | Code Llama | CodeLlama-70b-Python | 70B | 4096 | 40.29% | 55.49 | 59.14% | 36.07% | 41.06% | 7.59% | 0.00% | 42.03% | 43.04% | 40.76% | 32.46% | 41.78% | 41.00% | 10.96% | 19.50% |
| 49 | OctoPack | OctoGeeX | 6B | 8192 | 40.14% ± 1.55% | 42.28 | 62.54% | 37.84% | 26.39% | 15.67% | 2.20% | 42.24% | 33.23% | 46.02% | 39.10% | 39.85% | 39.96% | 20.90% | 31.11% |
| 50 | DeepSeek LLM | deepseek-llm-67b-base | 67B | 4096 | 39.87% | / | 57.15% | 48.73% | 32.53% | 9.17% | 4.17% | 35.50% | 43.17% | 46.15% | 34.40% | 39.98% | 36.00% | 30.00% | 24.44% |
| 51 | Llama 2 | Llama2-70B-Chat | 70B | 4096 | 39.30% | 29.98 | 56.95% | 38.02% | 33.71% | 7.96% | 7.64% | 35.65% | 42.87% | 42.56% | 36.11% | 40.89% | 34.40% | 22.44% | 28.14% |
| 52 | DeepSeek Coder | deepseek-coder-33b-base | 33B | 16384 | 38.75% | 52.45 | 56.73% | 44.55% | 19.85% | 14.95% | 8.33% | 33.36% | 43.73% | 46.06% | 31.23% | 43.99% | 25.50% | 14.49% | 28.02% |
| 53 | 01.AI | Yi-6B-Chat | 6B | 4096 | 38.14% ± 0.58% | / | 52.73% | 38.20% | 34.37% | 12.53% | 7.64% | 33.36% | 39.81% | 42.54% | 38.33% | 42.25% | 23.83% | 15.32% | 15.69% |
| 54 | Llama 2 | Llama2-70B | 70B | 4096 | 37.69% | 28.7 | 51.51% | 42.58% | 28.48% | 10.19% | 10.42% | 36.26% | 42.99% | 37.12% | 32.98% | 39.52% | 28.00% | 30.45% | 0.00% |
| 55 | Code Llama | CodeLlama-7b | 7B | 16384 | 37.62% ± 1.28% | 29.98 | 59.81% | 38.25% | 19.37% | 9.32% | 4.86% | 42.19% | 38.60% | 37.37% | 28.41% | 37.87% | 41.80% | 15.13% | 0.00% |
| 56 | Mistral Open | Mistral-7B-Instruct-v0.1 | 7B | 32768 | 37.55% ± 1.10% | / | 56.31% | 41.34% | 24.07% | 7.47% | 3.47% | 39.74% | 30.74% | 47.10% | 31.40% | 34.17% | 39.80% | 34.44% | 29.90% |
| 57 | InternLM | InternLM-Chat-20B | 20B | 16384 | 37.41% ± 0.75% | / | 56.31% | 32.30% | 20.40% | 18.44% | 7.06% | 45.38% | 34.67% | 34.25% | 31.63% | 34.51% | 41.60% | 18.18% | 23.51% |
| 58 | Qwen | Qwen-7B-Chat | 7B | 32768 | 37.36% ± 1.79% | 36 | 69.23% | 36.20% | 19.77% | 7.65% | 5.90% | 43.43% | 38.22% | 38.22% | 32.98% | 34.56% | 43.07% | 29.02% | 30.13% |
| 59 | DeepSeek LLM | deepseek-llm-7b-chat | 7B | 4096 | 36.75% ± 1.40% | / | 55.46% | 39.38% | 22.94% | 6.30% | 6.37% | 34.08% | 29.75% | 46.76% | 38.83% | 36.19% | 30.13% | 15.90% | 35.98% |
| 60 | Llama 2 | Llama2-7B-Chat | 7B | 4096 | 36.14% ± 1.05% | / | 54.17% | 35.35% | 24.72% | 9.44% | 9.03% | 35.53% | 33.29% | 39.16% | 37.51% | 37.64% | 28.50% | 21.35% | 27.76% |
| 61 | WizardLM | WizardCoder-3B-V1.0 | 3B | 2048 | 35.61% ± 0.42% | 32.92 | 57.44% | 35.61% | 15.23% | 11.30% | 6.60% | 39.25% | 32.08% | 41.34% | 26.96% | 35.83% | 35.40% | 19.25% | 26.50% |
| 62 | Code Llama | CodeLlama-7b-Instruct | 7B | 16384 | 35.15% ± 1.02% | 45.65 | 53.69% | 35.79% | 24.82% | 7.59% | 1.39% | 36.46% | 37.13% | 35.00% | 30.05% | 35.97% | 34.87% | 15.77% | 13.83% |
| 63 | StarCoder2 | 7B | 7B | 16384 | 34.90% ± 0.97% | 35.4 | 54.15% | 35.66% | 20.68% | 7.59% | 5.09% | 34.44% | 30.78% | 42.42% | 32.01% | 37.33% | 33.53% | 8.97% | 0.00% |
| 64 | InternLM | InternLM-Chat-7B | 7B | 8192 | 34.80% ± 0.90% | / | 55.80% | 32.39% | 20.76% | 12.70% | 1.85% | 35.31% | 34.30% | 39.75% | 28.52% | 35.23% | 34.57% | 17.65% | 16.86% |
| 65 | Baichuan2 | Baichuan2-13B-Chat | 13B | 4096 | 34.40% ± 1.34% | 19.5 | 53.77% | 27.69% | 24.19% | 6.85% | 14.12% | 37.03% | 35.93% | 36.39% | 24.88% | 34.62% | 31.07% | 22.63% | 18.28% |
| 66 | DeepSeek Coder | deepseek-coder-6.7b-base | 6.7B | 16384 | 33.66% ± 1.24% | 45.83 | 52.26% | 37.95% | 14.02% | 8.36% | 2.78% | 36.56% | 32.40% | 37.83% | 25.00% | 33.82% | 34.34% | 11.92% | 16.83% |
| 67 | Code Llama | CodeLlama-7b-Python | 7B | 16384 | 32.89% ± 0.45% | 40.48 | 51.02% | 28.69% | 24.53% | 7.59% | 6.94% | 30.38% | 38.54% | 32.73% | 29.81% | 35.27% | 30.40% | 8.97% | 11.31% |
| 68 | Llama 2 | Llama2-13B-Chat | 13B | 4096 | 32.29% ± 1.66% | / | 51.19% | 29.18% | 22.80% | 7.59% | 2.08% | 27.51% | 28.98% | 42.86% | 31.84% | 37.07% | 21.07% | 9.17% | 19.77% |
| 69 | WizardLM | WizardCoder-1B-V1.0 | 1B | 2048 | 31.94% ± 0.70% | 23.17 | 46.90% | 30.00% | 27.37% | 1.36% | 9.72% | 28.75% | 30.77% | 36.80% | 32.94% | 34.50% | 25.00% | 16.65% | 20.69% |
| 70 | Qwen | Qwen-7B | 7B | 32768 | 31.69% ± 0.29% | / | 52.65% | 32.18% | 15.18% | 2.10% | 1.85% | 33.78% | 30.71% | 36.78% | 22.83% | 31.09% | 34.07% | 15.71% | 17.12% |
| 71 | StarCoder2 | 3B | 3B | 16384 | 31.44% ± 1.92% | 31.7 | 46.65% | 35.17% | 18.91% | 8.70% | 3.94% | 29.01% | 35.23% | 33.79% | 26.96% | 36.13% | 26.13% | 3.72% | 0.00% |
| 72 | StarCoder | StarCode+ | 15.5B | 8192 | 30.67% ± 1.57% | / | 50.99% | 29.51% | 14.29% | 4.01% | 4.63% | 31.83% | 29.19% | 36.04% | 23.84% | 33.63% | 27.47% | 8.72% | 2.08% |
| 73 | StarCoder | StarCoder | 15.5B | 8192 | 30.66% ± 0.69% | 33.57 | 45.97% | 30.30% | 17.87% | 5.90% | 2.78% | 31.18% | 29.05% | 41.80% | 23.42% | 36.68% | 16.33% | 11.43% | 0.00% |
| 74 | CodeGen2.5 | CodeGen2.5-7B-Instruct | 7B | 2048 | 29.57% ± 1.53% | / | 50.36% | 22.01% | 20.25% | 6.67% | 6.46% | 28.76% | 25.66% | 37.70% | 25.77% | 32.35% | 24.76% | 11.54% | 0.00% |
| 75 | 🔒Mistral | mistral-tiny | ? | 32768 | 29.41% ± 0.26% | 28.7 | 52.53% | 20.42% | 14.60% | 7.28% | 4.17% | 33.32% | 27.59% | 32.89% | 20.69% | 28.31% | 29.67% | 18.78% | 38.00% |
| 76 | InternLM | InternLM-20B | 20B | 16384 | 29.41% ± 0.76% | / | 49.21% | 25.17% | 18.01% | 4.81% | 1.74% | 28.48% | 24.69% | 35.00% | 30.79% | 29.58% | 26.23% | 14.62% | 37.60% |
| 77 | DeepSeek Coder | deepseek-coder-5.7bmqa-base | 5.7B | 16384 | 28.92% ± 1.12% | / | 45.62% | 33.11% | 11.67% | 4.54% | 4.51% | 26.82% | 27.41% | 37.87% | 23.17% | 26.04% | 24.93% | 10.64% | 19.54% |
| 78 | ChatGLM | ChatGLM3-6B | 6B | 8192 | 28.23% ± 0.58% | 52.4 | 42.48% | 26.87% | 20.78% | 6.44% | 6.02% | 30.57% | 21.80% | 29.69% | 31.85% | 28.92% | 28.23% | 8.25% | 27.01% |
| 79 | Baichuan2 | Baichuan2-7B-Chat | 7B | 4096 | 27.53% ± 1.07% | 17.7 | 42.14% | 28.83% | 16.84% | 3.55% | 5.56% | 29.02% | 26.36% | 32.91% | 19.63% | 28.30% | 27.40% | 3.65% | 49.66% |
| 80 | gemma | gemma-2b-it | 2B | 8192 | 27.49% ± 0.52% | 17.7 | 43.43% | 29.73% | 13.99% | 0.62% | 5.56% | 22.98% | 26.43% | 37.85% | 23.49% | 29.17% | 20.57% | 9.08% | 31.81% |
| 81 | Qwen | Qwen-1.8B-Chat | 1.8B | 32768 | 26.84% ± 1.08% | / | 40.35% | 25.15% | 22.50% | 5.25% | 1.23% | 25.96% | 27.23% | 26.91% | 24.18% | 29.00% | 25.11% | 5.81% | 19.65% |
| 82 | DeepSeek MoE | deepseek-moe-16b-base | 16B / 2.8B | 16384 | 26.65% ± 0.97% | / | 41.68% | 31.71% | 12.27% | 4.21% | 0.00% | 28.09% | 25.69% | 31.15% | 19.66% | 27.77% | 27.11% | 5.38% | 22.26% |
| 83 | 01.AI | Yi-9B | 9B | 4096 | 26.39% ± 0.42% | 39 | 41.18% | 29.89% | 14.57% | 3.33% | 0.00% | 20.83% | 27.06% | 34.48% | 24.58% | 30.21% | 17.00% | 5.96% | 14.34% |
| 84 | Baichuan2 | Baichuan2-13B-Base | 13B | 4096 | 26.32% ± 1.23% | / | 43.01% | 21.48% | 16.87% | 6.47% | 4.98% | 22.46% | 26.54% | 31.79% | 25.63% | 30.05% | 16.24% | 9.23% | 13.52% |
| 85 | DeepSeek LLM | deepseek-llm-7b-base | 7B | 4096 | 25.34% ± 1.08% | 26.2 | 36.58% | 30.59% | 15.33% | 2.01% | 5.56% | 19.59% | 27.42% | 29.23% | 27.22% | 28.67% | 15.00% | 8.97% | 25.29% |
| 86 | Llama 2 | Llama2-13B | 13B | 4096 | 24.50% ± 0.73% | / | 38.09% | 25.73% | 15.00% | 6.48% | 0.00% | 21.25% | 25.09% | 28.38% | 24.29% | 26.79% | 19.80% | 9.68% | 4.62% |
| 87 | Baichuan2 | Baichuan2-7B-Base | 7B | 4096 | 23.50% ± 1.56% | / | 36.59% | 23.93% | 13.01% | 5.99% | 4.17% | 21.05% | 22.93% | 28.86% | 21.49% | 26.03% | 19.33% | 4.68% | 10.70% |
| 88 | DeepSeek Coder | deepseek-coder-1.3b-base | 1.3B | 16384 | 23.17% ± 1.47% | 32.13 | 37.06% | 26.74% | 8.03% | 2.84% | 4.17% | 16.05% | 20.93% | 34.16% | 24.68% | 27.02% | 14.40% | 4.68% | 24.92% |
| 89 | Qwen | Qwen-1.8B | 1.8B | 32768 | 23.12% ± 1.13% | / | 37.81% | 18.94% | 10.04% | 3.70% | 3.70% | 18.04% | 20.99% | 29.28% | 16.60% | 25.70% | 18.40% | 3.72% | 0.00% |
| 90 | Mistral Open | Mistral-7B-v0.1 | 7B | 32768 | 22.72% ± 1.51% | 28.7 | 34.86% | 23.00% | 15.83% | 6.30% | 0.00% | 20.01% | 25.52% | 24.24% | 21.32% | 25.01% | 17.47% | 10.32% | 0.00% |
| 91 | Llama 2 | Llama2-7B | 7B | 4096 | 22.35% ± 1.70% | 14.6 | 37.45% | 21.33% | 10.00% | 1.85% | 4.17% | 20.57% | 18.80% | 28.69% | 22.51% | 25.28% | 18.27% | 0.77% | 12.64% |
| 92 | 01.AI | Yi-34B | 34B | 4096 | 22.01% | / | 34.64% | 26.46% | 7.73% | 1.85% | 4.17% | 23.15% | 16.96% | 31.36% | 15.32% | 23.10% | 22.40% | 6.15% | 11.46% |
| 93 | 🔒davinci | davinci-002 | ? | 16384 | 21.25% ± 1.17% | / | 33.66% | 19.26% | 13.61% | 5.27% | 3.70% | 15.05% | 19.63% | 30.35% | 22.70% | 25.42% | 13.36% | 2.61% | 4.33% |
| 94 | Mistral Open | mixtral-8x7B | 46.7B / 12.9B | 32768 | 21.21% | / | 32.76% | 20.54% | 15.23% | 3.70% | 2.08% | 18.04% | 14.02% | 32.51% | 22.78% | 23.57% | 13.50% | 10.00% | 16.03% |
| 95 | Phi | Phi1.5 | 1.5B | 2048 | 20.56% ± 0.09% | / | 32.15% | 20.61% | 14.27% | 3.40% | 0.00% | 21.04% | 22.86% | 21.15% | 15.53% | 21.83% | 21.80% | 1.92% | 13.97% |
| 96 | 01.AI | Yi-6B | 6B | 4096 | 19.93% ± 1.24% | / | 31.84% | 18.91% | 13.13% | 0.99% | 2.78% | 13.75% | 23.72% | 23.54% | 20.37% | 23.42% | 14.58% | 0.00% | 4.54% |
| 97 | CodeGeeX | CodeGeeX2-6B | 6B | 8192 | 19.88% ± 0.36% | 33.49 | 31.40% | 17.41% | 14.02% | 2.22% | 1.39% | 19.90% | 19.99% | 25.39% | 14.44% | 22.08% | 16.11% | 4.10% | 9.78% |
| 98 | CodeGen2 | CodeGen2-16B | 16B | 2048 | 16.97% ± 1.15% | / | 27.46% | 18.30% | 8.08% | 1.23% | 1.39% | 13.00% | 17.28% | 24.04% | 14.23% | 20.77% | 7.58% | 7.05% | 0.00% |
| 99 | Phi | Phi2 | 1.3B | 2048 | 16.74% ± 0.64% | 48.2 | 28.96% | 13.97% | 8.23% | 5.12% | 0.00% | 17.45% | 14.49% | 17.17% | 18.28% | 18.62% | 13.33% | 1.73% | 18.18% |
| 100 | InternLM | InternLM-7B | 7B | 8192 | 16.26% ± 2.21% | / | 25.17% | 14.34% | 10.86% | 2.59% | 6.25% | 8.48% | 16.95% | 30.14% | 10.71% | 20.19% | 4.89% | 1.92% | 24.80% |
| 101 | gemma | gemma-7b | 7B | 8192 | 16.05% ± 0.80% | 35.4 | 24.14% | 14.96% | 7.53% | 2.65% | 0.00% | 6.98% | 15.79% | 30.06% | 14.05% | 19.73% | 6.44% | 2.56% | 8.45% |
| 102 | IEITYuan | Yuan2-51B-hf | 51B | 4096 | 15.25% | / | 25.61% | 12.20% | 6.06% | 2.78% | 8.33% | 20.16% | 16.37% | 15.38% | 4.76% | 15.06% | 16.83% | 1.92% | 29.55% |
| 103 | gemma | gemma-2b | 2B | 8192 | 14.62% ± 0.50% | 25 | 23.18% | 13.23% | 10.53% | 4.07% | 0.00% | 12.16% | 12.89% | 24.70% | 6.33% | 16.89% | 11.33% | 0.00% | 0.00% |
| 104 | Phi | Phi1 | 2.7B | 2048 | 14.28% ± 0.99% | 51.22 | 20.78% | 17.23% | 8.08% | 5.87% | 0.00% | 8.26% | 18.93% | 18.09% | 12.91% | 18.04% | 3.33% | 1.28% | 26.65% |
| 105 | CodeGen | CodeGen-16B-multi | 16B | 2048 | 13.62% ± 1.18% | 19.26 | 20.79% | 13.19% | 6.82% | 1.85% | 0.00% | 12.17% | 9.45% | 15.71% | 5.24% | 16.44% | 5.00% | 0.00% | 19.11% |
| 106 | IEITYuan | Yuan2-102B-hf | 102B | 4096 | 10.48% | / | 18.18% | 7.77% | 6.32% | 1.85% | 0.00% | 10.84% | 8.29% | 14.36% | 3.46% | 8.41% | 18.33% | 0.00% | 6.31% |
| 107 | IEITYuan | Yuan2-2B-hf | 2B | 8192 | 7.28% ± 1.01% | / | 9.11% | 8.11% | 5.56% | 5.56% | 2.78% | 4.01% | 8.29% | 10.28% | 7.62% | 8.80% | 4.27% | 0.00% | 6.31% |
| | Mean | | | | 38.08% | | 57.50% | 38.76% | 25.13% | 10.52% | 5.68% | 39.33% | 36.29% | 42.22% | 33.15% | 38.20% | 38.06% | 20.67% | 23.92% |
| | Human | 10 Highest-Voted Answer Posts | / | | 65.18% | / | 67.56% | 59.09% | 72.73% | 53.87% | 64.58% | 29.73% | 83.28% | 77.33% | 84.09% | 83.27% | 7.00% | 30.38% | 79.94% |
| | | Highest-Voted Answer Post | / | | 56.28% | / | 58.78% | 51.82% | 61.36% | 48.31% | 53.47% | 25.00% | 72.16% | 69.55% | 70.20% | 73.01% | 6.00% | 16.92% | 79.94% |
| | | Officially-Accepted Answer Post | / | | 52.90% | / | 56.63% | 49.55% | 53.03% | 42.76% | 53.47% | 27.03% | 62.24% | 64.70% | 69.01% | 67.58% | 6.00% | 21.73% | 79.94% |

Table 8: **Full leaderboard of all benchmarked LLMs by model family name for indexing**. Same content as Table 7. Evaluation protocol in Section 3 and details explained in Appendix E. Icon "🔒" stands for proprietary models otherwise open-source. As a reference, HumanEval scores digested from [26] and each model's report are shown. Bar colors stand for General Base , General Finetuned , Code Base , and Code Finetuned models respectively. Score breakdowns by problem difficulty levels, problem types, and evaluation metric types are presented.

| No | Model Family | Model Name | Size (# Param.) | Context Length | InfiBench Score | HumanEval | Difficulty Levels | | | | | Problem Type | | | | Evaluation Metric Type | | | |
|---|---|---|---|---|---|---|---|---|---|---|---|---|---|---|---|---|---|---|---|
| | | | | | | | Level 1 | Level 2 | Level 3 | Level 4 | Level 5 | Code Completion | Code Debugging | Knowledge QA | Config & Env Debugging | Keyword Matching | Unit Testing | Blank Filling | Dialogue Similarity |
| 1 | 01.AI | Yi-34B-Chat | 34B | 4096 | 49.58% | / | 76.81% | 47.15% | 29.32% | 26.39% | 4.17% | 44.10% | 44.75% | 62.29% | 49.84% | 53.14% | 35.40% | 36.15% | 33.07% |
| 2 | 01.AI | Yi-6B-Chat | 6B | 4096 | 38.14% ± 0.58% | / | 52.73% | 38.20% | 34.37% | 12.53% | 7.64% | 33.36% | 39.81% | 42.54% | 38.33% | 43.26% | 23.83% | 15.32% | 15.69% |
| 3 | 01.AI | Yi-9B | 9B | 4096 | 26.39% ± 0.42% | 39 | 41.18% | 29.89% | 14.57% | 3.33% | 0.00% | 20.83% | 27.06% | 34.48% | 24.58% | 30.21% | 17.60% | 5.96% | 14.34% |
| 4 | 01.AI | Yi-34B | 34B | 4096 | 22.01% | / | 34.64% | 26.46% | 7.73% | 1.85% | 4.17% | 23.15% | 16.96% | 31.36% | 15.32% | 23.10% | 22.40% | 6.15% | 11.46% |
| 5 | 01.AI | Yi-6B | 6B | 4096 | 19.93% ± 1.24% | / | 31.84% | 18.91% | 13.13% | 0.99% | 2.78% | 13.75% | 23.72% | 23.54% | 20.37% | 23.42% | 14.58% | 0.00% | 4.54% |
| 6 | Baichuan2 | Baichuan2-13B-Chat | 13B | 4096 | 34.40% ± 1.34% | 19.5 | 53.77% | 27.69% | 24.19% | 6.85% | 14.12% | 37.03% | 35.93% | 36.39% | 24.88% | 34.62% | 31.07% | 22.63% | 18.28% |
| 7 | Baichuan2 | Baichuan2-7B-Chat | 7B | 4096 | 27.53% ± 1.07% | 17.7 | 42.14% | 28.83% | 16.84% | 3.55% | 5.56% | 29.02% | 26.36% | 32.91% | 19.63% | 28.30% | 27.40% | 3.65% | 49.66% |
| 8 | Baichuan2 | Baichuan2-13B-Base | 13B | 4096 | 26.32% ± 1.23% | / | 43.01% | 21.48% | 16.87% | 6.47% | 4.98% | 22.46% | 26.54% | 31.79% | 25.63% | 30.05% | 16.24% | 9.23% | 13.52% |
| 9 | Baichuan2 | Baichuan2-7B-Base | 7B | 4096 | 23.50% ± 1.56% | / | 36.99% | 23.93% | 13.01% | 5.99% | 4.17% | 21.05% | 22.93% | 28.98% | 21.49% | 26.03% | 19.33% | 4.68% | 10.70% |
| 10 | ChatGLM | ChatGLM3-6B | 6B | 8192 | 28.23% ± 0.8% | 52.4 | 42.48% | 26.87% | 20.78% | 6.64% | 6.02% | 30.57% | 21.80% | 29.69% | 31.85% | 28.92% | 28.23% | 8.25% | 27.01% |
| 11 | 🔒Claude 3 | Claude 3 Opus | ? | 200000 | 63.89% | 73 | 84.36% | 78.95% | 39.98% | 31.76% | 18.06% | 65.18% | 62.94% | 65.86% | 60.49% | 60.07% | 61.80% | 59.36% | 44.91% |
| 12 | 🔒Claude 3 | Claude 3 Sonnet | ? | 200000 | 58.20% | 84.9 | 80.15% | 65.55% | 42.48% | 18.06% | 15.28% | 62.61% | 52.34% | 63.61% | 52.12% | 54.22% | 66.00% | 46.35% | 25.62% |
| 13 | 🔒Claude 3 | Claude 3 Haiku | ? | 200000 | 57.57% | 75.9 | 79.86% | 66.06% | 40.23% | 21.76% | 10.42% | 61.71% | 48.68% | 62.85% | 56.71% | 55.78% | 58.40% | 44.62% | 36.40% |
| 14 | Code Llama | CodeLlama-34b-Instruct | 34B | 16384 | 50.45% | 50.79 | 72.60% | 55.07% | 33.16% | 18.43% | 9.72% | 51.71% | 48.37% | 61.36% | 37.04% | 48.14% | 51.20% | 47.76% | 28.55% |
| 15 | Code Llama | CodeLlama-34b | 34B | 16384 | 47.36% | 45.11 | 72.07% | 43.34% | 29.32% | 21.20% | 13.54% | 53.74% | 50.09% | 51.52% | 26.59% | 43.18% | 57.33% | 37.37% | 24.85% |
| 16 | Code Llama | CodeLlama-13b-Instruct | 13B | 16384 | 46.37% ± 1.24% | 50.6 | 69.07% | 45.99% | 34.37% | 11.42% | 7.52% | 48.65% | 45.18% | 49.67% | 39.83% | 47.71% | 50.47% | 20.90% | 12.45% |
| 17 | Code Llama | CodeLlama-34b-Python | 34B | 16384 | 43.13% | 53.29 | 66.02% | 40.76% | 36.06% | 6.94% | 0.00% | 50.14% | 40.48% | 43.64% | 34.13% | 40.40% | 51.00% | 27.63% | 16.67% |
| 18 | Code Llama | CodeLlama-70b-Instruct | 70B | 4096 | 42.82% | 75.6 | 59.09% | 44.14% | 38.48% | 12.22% | 7.64% | 38.20% | 44.99% | 46.87% | 42.38% | 48.34% | 32.00% | 16.09% | 5.62% |
| 19 | Code Llama | CodeLlama-13b | 13B | 16384 | 41.66% ± 0.84% | 35.07 | 62.77% | 40.40% | 31.11% | 7.99% | 7.41% | 38.17% | 44.56% | 43.00% | 41.72% | 45.44% | 54.80% | 14.79% | 2.47% |
| 20 | Code Llama | CodeLlama-13b-Python | 13B | 16384 | 41.31% ± 0.90% | 42.89 | 62.93% | 40.80% | 28.61% | 10.37% | 5.21% | 49.95% | 44.60% | 36.68% | 27.22% | 40.58% | 51.07% | 11.92% | 13.64% |
| 21 | Code Llama | CodeLlama-70b | 70B | 4096 | 40.60% | 55.5 | 60.97% | 37.42% | 35.68% | 7.59% | 4.17% | 47.18% | 39.10% | 39.09% | 33.21% | 40.54% | 45.00% | 19.23% | 8.56% |
| 22 | Code Llama | CodeLlama-70b-Python | 70B | 4096 | 40.29% | 55.49 | 59.14% | 36.07% | 41.06% | 7.59% | 0.00% | 42.03% | 43.04% | 40.76% | 32.46% | 41.78% | 41.00% | 10.96% | 19.50% |
| 23 | Code Llama | CodeLlama-7b | 7B | 16384 | 37.62% ± 1.28% | 29.98 | 59.81% | 38.25% | 19.37% | 9.32% | 4.86% | 42.19% | 38.60% | 37.37% | 28.41% | 37.87% | 41.80% | 15.13% | 0.00% |
| 24 | Code Llama | CodeLlama-7b-Instruct | 7B | 16384 | 35.15% ± 1.02% | 45.65 | 53.69% | 35.79% | 24.82% | 7.59% | 1.39% | 36.46% | 37.13% | 35.00% | 30.05% | 35.97% | 34.87% | 15.77% | 13.83% |
| 25 | Code Llama | CodeLlama-7b-Python | 7B | 16384 | 32.89% ± 0.45% | 40.48 | 51.25% | 28.69% | 24.32% | 7.59% | 6.94% | 30.38% | 38.34% | 32.37% | 29.81% | 35.27% | 30.40% | 8.97% | 11.31% |
| 26 | CodeGeeX | CodeGeeX2-6B | 6B | 8192 | 19.88% ± 0.36% | 33.49 | 31.40% | 17.14% | 14.02% | 2.22% | 4.86% | 18.78% | 19.97% | 25.39% | 14.44% | 22.08% | 16.11% | 4.10% | 9.78% |
| 27 | CodeGen | CodeGen-16B-multi | 16B | 2048 | 13.62% ± 1.18% | 19.26 | 20.79% | 13.19% | 10.86% | 2.84% | 0.00% | 11.36% | 17.90% | 13.45% | 11.43% | 16.44% | 6.90% | 3.08% | 8.77% |
| 28 | CodeGen2 | CodeGen2-16B | 16B | 2048 | 16.97% ± 1.15% | / | 27.46% | 18.30% | 8.08% | 1.23% | 1.39% | 13.00% | 17.28% | 24.04% | 14.23% | 20.77% | 7.45% | 7.05% | 0.00% |
| 29 | CodeGen2.5 | CodeGen2.5-7B-Instruct | 7B | 2048 | 29.57% ± 1.53% | / | 50.36% | 22.07% | 20.25% | 6.67% | 0.46% | 28.76% | 25.96% | 37.70% | 25.77% | 32.35% | 24.76% | 11.54% | 0.00% |
| 30 | 🔒davinci | davinci-002 | ? | / | 21.25% ± 1.17% | / | 33.66% | 19.26% | 13.61% | 5.27% | 3.70% | 15.05% | 19.63% | 30.35% | 22.70% | 25.42% | 13.36% | 2.61% | 4.33% |
| 31 | DeepSeek Coder | deepSeek-coder-V2-instruct | 236B / 21B | 128000 | 65.49% | 90.2 | 88.77% | 76.97% | 50.58% | 17.31% | 12.50% | 74.77% | 63.89% | 59.57% | 59.30% | 58.91% | 76.00% | 55.77% | 37.34% |
| 32 | DeepSeek Coder | deepseek-coder-33b-instruct | 33B | 16384 | 62.96% | 80.02 | 87.58% | 72.02% | 44.12% | 15.83% | 16.67% | 71.26% | 57.14% | 63.14% | 56.81% | 59.01% | 77.00% | 30.00% | 36.09% |
| 33 | DeepSeek Coder | deepseek-coder-6.7b-instruct | 6.7B | 16384 | 53.25% ± 0.40% | 80.22 | 77.88% | 56.30% | 35.18% | 18.89% | 9.72% | 65.95% | 46.44% | 52.46% | 42.12% | 48.24% | 70.40% | 26.90% | 23.48% |
| 34 | DeepSeek Coder | deepseek-coder-1.3b-instruct | 1.3B | 16384 | 41.32% ± 1.12% | 64.6 | 65.48% | 41.42% | 25.48% | 6.30% | 2.78% | 41.91% | 42.66% | 36.38% | 41.40% | 42.20% | 46.52% | 16.52% | 24.32% |
| 35 | DeepSeek Coder | deepseek-coder-33b-base | 33B | 16384 | 38.75% | 52.45 | 76.73% | 44.55% | 19.85% | 14.95% | 8.33% | 33.36% | 43.73% | 46.06% | 31.23% | 43.99% | 25.50% | 14.49% | 28.02% |
| 36 | DeepSeek Coder | deepseek-coder-6.7b-base | 6.7B | 16384 | 33.66% ± 1.24% | 45.83 | 53.26% | 37.95% | 14.02% | 8.56% | 2.78% | 36.56% | 32.40% | 37.83% | 25.00% | 35.17% | 33.47% | 11.92% | 8.81% |
| 37 | DeepSeek Coder | deepseek-coder-5.7bmqa-base | 5.7B | 16384 | 28.92% ± 1.12% | / | 45.62% | 33.11% | 11.67% | 4.54% | 4.51% | 26.82% | 27.41% | 37.87% | 23.17% | 36.64% | 24.93% | 10.64% | 19.54% |
| 38 | DeepSeek Coder | deepseek-coder-1.3b-base | 1.3B | 16384 | 23.17% ± 1.47% | 32.13 | 37.06% | 26.74% | 8.03% | 2.84% | 4.17% | 16.05% | 20.93% | 34.16% | 24.68% | 27.02% | 14.40% | 4.68% | 24.92% |
| 39 | DeepSeek LLM | deepseek-llm-67b-chat | 67B | 4096 | 57.41% | / | 82.96% | 63.03% | 39.09% | 22.60% | 5.21% | 61.42% | 52.73% | 58.72% | 55.63% | 53.14% | 63.00% | 51.41% | 36.68% |
| 40 | DeepSeek LLM | deepseek-llm-67b-base | 67B | 4096 | 39.87% | 42.7 | 57.15% | 48.73% | 24.32% | 9.17% | 4.17% | 35.50% | 43.17% | 46.15% | 34.40% | 39.98% | 36.00% | 30.00% | 22.46% |
| 41 | DeepSeek LLM | deepseek-llm-7b-chat | 7B | 4096 | 36.75% ± 1.28% | / | 55.46% | 39.38% | 22.94% | 6.30% | 6.37% | 34.08% | 29.75% | 46.76% | 38.83% | 39.15% | 30.13% | 15.90% | 35.98% |
| 42 | DeepSeek LLM | deepseek-llm-7b-base | 7B | 4096 | 25.34% ± 1.08% | 26.2 | 36.58% | 30.59% | 15.33% | 2.01% | 5.56% | 19.59% | 27.42% | 29.23% | 27.22% | 28.67% | 15.00% | 8.97% | 29.55% |
| 43 | DeepSeek MoE | deepseek-moe-16b-chat | 16B / 2.8B | 16384 | 45.18% ± 1.65% | / | 68.15% | 46.72% | 27.55% | 10.17% | 11.23% | 47.19% | 46.54% | 45.58% | 39.09% | 45.15% | 44.73% | 25.85% | 20.70% |
| 44 | DeepSeek MoE | deepseek-moe-16b-base | 16B / 2.8B | 16384 | 26.65% ± 0.97% | / | 41.68% | 31.71% | 12.27% | 4.21% | 0.00% | 28.09% | 25.69% | 31.15% | 19.66% | 25.77% | 27.11% | 5.38% | 22.26% |
| 45 | gemma | gemma-7b-it | 7B | 8192 | 40.68% ± 1.23% | 28.7 | 60.94% | 42.94% | 28.86% | 5.75% | 4.86% | 42.60% | 36.37% | 47.75% | 34.52% | 40.68% | 41.40% | 19.04% | 30.44% |
| 46 | gemma | gemma-2b-it | 2B | 8192 | 27.49% ± 0.52% | 17.7 | 43.43% | 29.73% | 13.99% | 0.62% | 5.56% | 22.98% | 26.43% | 37.85% | 23.49% | 29.17% | 20.75% | 9.08% | 31.81% |
| 47 | gemma | gemma-7b | 7B | 8192 | 16.05% ± 0.80% | 35.4 | 27.46% | 14.96% | 7.53% | 2.65% | 0.00% | 6.98% | 15.79% | 30.06% | 14.05% | 19.73% | 6.44% | 2.56% | 8.45% |
| 48 | gemma | gemma-2b | 2B | 8192 | 14.62% ± 0.50% | 25 | 23.18% | 13.23% | 10.53% | 4.07% | 0.00% | 12.16% | 12.89% | 24.70% | 8.33% | 16.99% | 11.33% | 0.00% | 0.00% |
| 49 | 🔒GPT-3.5 | GPT-3.5-turbo-0613 | ? | 4096 | 56.47% ± 1.34% | 72.6 | 93.08% | 49.77% | 31.36% | 14.30% | 7.64% | 64.91% | 48.50% | 59.47% | 49.64% | 51.28% | 70.07% | 40.90% | 40.13% |
| 50 | 🔒GPT-4 | GPT-4-0613 | ? | 8192 | 70.64% ± 0.82% | 88.4 | 92.31% | 62.48% | 51.90% | 31.91% | 0.00% | 75.23% | 69.74% | 68.55% | 66.61% | 66.61% | 76.00% | 58.08% | 84.27% |
| 51 | 🔒GPT-4 | GPT-4-turbo-1106 | ? | 8192 | 68.42% ± 0.38% | 85.4 | 89.90% | 78.57% | 54.16% | 30.91% | 16.20% | 74.82% | 65.36% | 67.47% | 62.95% | 64.98% | 76.40% | 53.91% | 52.85% |
| 52 | 🔒GPT-4 | GPT-4o-2024-05-13 | ? | 8192 | 66.19% | 90.2 | 91.29% | 78.66% | 46.43% | 28.05% | 5.21% | 75.00% | 59.32% | 65.65% | 61.70% | 61.59% | 76.00% | 49.74% | 70.73% |
| 53 | IEITYuan | Yuan2-51B-hf | 51B | 4096 | 15.25% | / | 25.61% | 12.20% | 6.06% | 2.78% | 8.33% | 20.16% | 16.37% | 15.38% | 4.76% | 15.09% | 16.83% | 1.92% | 29.55% |
| 54 | IEITYuan | Yuan2-102B-hf | 102B | 4096 | 10.48% | / | 18.18% | 7.77% | 6.82% | 1.85% | 0.00% | 17.12% | 9.45% | 6.71% | 5.24% | 8.41% | 18.33% | 0.00% | 19.11% |
| 55 | IEITYuan | Yuan2-2B-hf | 2B | 8192 | 7.28% ± 1.01% | / | 9.11% | 8.11% | 5.56% | 5.56% | 2.78% | 4.01% | 8.29% | 10.28% | 7.62% | 8.80% | 4.27% | 0.00% | 6.31% |
| 56 | InternLM | InternLM-Chat-20B | 20B | 16384 | 37.41% ± 0.75% | / | 59.98% | 32.30% | 20.40% | 18.44% | 7.06% | 45.38% | 34.67% | 34.25% | 31.63% | 34.51% | 46.20% | 18.18% | 23.51% |
| 57 | InternLM | InternLM-Chat-7B | 7B | 8192 | 34.80% ± 0.90% | / | 55.80% | 32.30% | 20.76% | 12.70% | 1.85% | 35.31% | 34.30% | 39.75% | 28.52% | 35.23% | 34.57% | 17.65% | 16.86% |
| 58 | InternLM | InternLM-20B | 20B | 16384 | 29.41% ± 0.76% | / | 49.21% | 25.17% | 18.01% | 4.81% | 1.74% | 28.39% | 26.30% | 30.79% | 29.58% | 30.30% | 29.58% | 26.24% | 37.60% |
| 59 | InternLM | InternLM-7B | 7B | 8192 | 16.26% ± 2.21% | / | 25.17% | 14.34% | 10.86% | 2.59% | 6.25% | 8.48% | 16.95% | 30.14% | 10.71% | 20.19% | 4.89% | 1.92% | 24.80% |
| 60 | Llama 2 | Llama2-70B-Chat | 70B | 4096 | 39.30% | / | 56.95% | 38.02% | 33.71% | 7.96% | 7.64% | 35.65% | 42.87% | 42.56% | 36.11% | 40.89% | 34.40% | 22.44% | 28.14% |
| 61 | Llama 2 | Llama2-70B | 70B | 4096 | 37.69% | 28.7 | 51.51% | 42.58% | 28.48% | 10.19% | 10.42% | 36.26% | 42.99% | 37.12% | 32.98% | 39.52% | 28.00% | 30.45% | 0.00% |
| 62 | Llama 2 | Llama2-7B-Chat | 7B | 4096 | 36.14% ± 1.05% | / | 54.17% | 35.35% | 24.72% | 9.44% | 9.03% | 35.53% | 33.29% | 39.16% | 37.51% | 37.64% | 28.50% | 21.35% | 27.76% |
| 63 | Llama 2 | Llama2-13B-Chat | 13B | 4096 | 32.29% ± 1.66% | / | 51.19% | 29.18% | 22.80% | 7.59% | 2.08% | 27.51% | 28.98% | 42.86% | 31.84% | 37.07% | 21.07% | 9.17% | 19.77% |
| 64 | Llama 2 | Llama2-13B | 13B | 4096 | 24.50% ± 0.73% | / | 38.09% | 25.73% | 15.00% | 6.48% | 0.00% | 21.25% | 25.09% | 28.38% | 24.29% | 26.79% | 19.80% | 9.68% | 4.62% |
| 65 | Llama 2 | Llama2-7B | 7B | 4096 | 22.35% ± 1.70% | 14.6 | 37.45% | 21.33% | 10.00% | 1.85% | 4.17% | 20.57% | 18.80% | 28.69% | 22.51% | 25.28% | 18.27% | 0.77% | 12.64% |
| 66 | Magicoder | Magicoder-S-CL-7B | 7B | 16384 | 52.71% ± 0.72% | 70.7 | 79.97% | 50.42% | 40.20% | 13.45% | 12.50% | 51.39% | 51.98% | 56.97% | 50.58% | 53.28% | 56.67% | 21.41% | 26.97% |
| 67 | Magicoder | Magicoder-S-DS-6.7B | 6.7B | 16384 | 51.46% ± 1.09% | 76.8 | 78.93% | 51.02% | 28.91% | 15.93% | 6.48% | 62.54% | 46.45% | 55.74% | 33.84% | 45.64% | 69.13% | 31.45% | 27.86% |
| 68 | Magicoder | Magicoder-DS-6.7B | 6.7B | 16384 | 43.47% ± 0.21% | / | 67.04% | 48.33% | 23.11% | 13.64% | 0.69% | 52.73% | 40.42% | 48.14% | 25.61% | 38.37% | 56.73% | 29.81% | 38.07% |
| 69 | Magicoder | Magicoder-CL-7B | 7B | 16384 | 41.71% ± 0.76% | / | 70.38% | 36.48% | 23.06% | 10.33% | 0.35% | 49.26% | 35.11% | 45.41% | 33.47% | 37.85% | 52.27% | 19.91% | 39.21% |
| 70 | 🔒Mistral | mistral-large | ? | 32768 | 58.22% | 69.5 | 81.76% | 66.59% | 41.66% | 23.62% | 4.17% | 66.69% | 50.10% | 60.21% | 53.24% | 53.17% | 67.00% | 45.64% | 42.66% |
| 71 | 🔒Mistral | mistral-small | ? | 32768 | 55.62% ± 0.46% | / | 82.98% | 55.98% | 35.72% | 22.58% | 10.07% | 63.56% | 44.12% | 64.13% | 47.75% | 50.56% | 68.00% | 39.08% | 53.32% |
| 72 | 🔒Mistral | mistral-medium | ? | 32768 | 40.95% ± 0.41% | / | 72.59% | 30.34% | 19.14% | 8.15% | 7.29% | 41.49% | 34.39% | 49.19% | 39.09% | 38.54% | 42.67% | 33.85% | 18.26% |
| 73 | 🔒Mistral | mistral-tiny | ? | 32768 | 29.41% ± 0.26% | 28.7 | 52.53% | 20.42% | 14.60% | 7.28% | 4.17% | 33.32% | 27.59% | 32.89% | 20.69% | 28.31% | 29.67% | 18.78% | 38.00% |
| 74 | Mistral Open | Codestral-22b | 22B | 32768 | 62.98% ± 0.56% | 81.1 | 88.64% | 69.90% | 49.97% | 17.11% | 5.90% | 68.75% | 63.65% | 61.07% | 57.72% | 57.72% | 73.33% | 49.52% | 57.08% |
| 75 | Mistral Open | Mistral-7B-Instruct | 46.7B / 12.9B | 32768 | 55.55% | 37.8 | 82.19% | 56.72% | 31.53% | 24.00% | 17.36% | 54.01% | 51.57% | 63.69% | 53.99% | 56.14% | 50.40% | 35.58% | 61.75% |
| 76 | Mistral Open | Mistral-7B-Instruct-v0.1 | 7B | 32768 | 37.55% ± 1.10% | / | 56.31% | 41.34% | 24.07% | 7.47% | 3.47% | 39.74% | 30.74% | 47.10% | 31.40% | 34.17% | 39.80% | 34.44% | 29.90% |
| 77 | Mistral Open | Mistral-7B-v0.1 | 7B | 32768 | 22.72% ± 1.51% | 28.7 | 34.86% | 23.00% | 15.83% | 6.30% | 0.00% | 20.01% | 25.52% | 24.24% | 21.32% | 25.01% | 17.47% | 10.32% | 0.00% |
| 78 | Mistral Open | mixtral-8x7B | 46.7B / 12.9B | 32768 | 21.21% | / | 32.76% | 20.54% | 15.23% | 3.70% | 2.08% | 18.04% | 14.02% | 32.51% | 22.78% | 23.57% | 13.50% | 10.00% | 16.03% |
| 79 | OctoPack | OctoCoder | 15.5B | 8192 | 44.55% ± 0.79% | 45.3 | 68.19% | 41.61% | 29.39% | 12.96% | 11.11% | 46.56% | 37.62% | 53.57% | 39.56% | 44.18% | 47.07% | 20.09% | 39.20% |
| 80 | OctoPack | OctoGeeX | 6B | 8192 | 40.14% ± 1.55% | 42.28 | 62.54% | 37.84% | 26.39% | 15.67% | 2.20% | 42.24% | 33.23% | 46.00% | 39.10% | 39.85% | 39.96% | 20.90% | 31.11% |
| 81 | Phi | Phi1.5 | 1.5B | 2048 | 20.56% ± 0.09% | / | 32.15% | 20.61% | 14.27% | 3.40% | 0.00% | 21.04% | 22.86% | 21.15% | 15.53% | 19.85% | 21.80% | 1.92% | 13.97% |
| 82 | Phi | Phi2 | 2.7B | 2048 | 16.74% ± 0.64% | 48.2 | 28.96% | 13.97% | 8.23% | 5.12% | 0.00% | 17.45% | 14.49% | 17.17% | 18.28% | 18.62% | 13.33% | 1.73% | 18.18% |
| 83 | Phi | Phi1 | 2.7B | 2048 | 14.28% ± 0.99% | 51.22 | 20.70% | 17.23% | 8.08% | 5.87% | 0.00% | 8.26% | 18.93% | 18.09% | 12.91% | 18.04% | 3.33% | 1.28% | 26.65% |
| 84 | Phind | Phind-CodeLlama-34B-v2 | 34B | 4096 | 59.00% | 71.95 | 83.67% | 55.57% | 53.12% | 15.09% | 14.93% | 58.24% | 58.30% | 63.60% | 55.33% | 59.63% | 58.40% | 35.26% | 24.19% |
| 85 | Phind | Phind-CodeLlama-34B-v1 | 34B | 4096 | 58.47% | 65.85 | 81.38% | 63.85% | 47.05% | 22.63% | 5.21% | 66.13% | 56.94% | 56.79% | 49.48% | 55.71% | 66.00% | 38.78% | 35.39% |
| 86 | Phind | Phind-CodeLlama-34B-Python-v1 | 34B | 4096 | 52.17% | 70.22 | 80.54% | 48.44% | 42.58% | 8.57% | 1.04% | 54.41% | 52.34% | 57.11% | 41.47% | 51.04% | 57.80% | 27.18% | 39.76% |
| 87 | Qwen | Qwen-72B | 72B | 32768 | 55.34% | / | 81.98% | 57.40% | 41.61% | 13.24% | 4.17% | 61.06% | 53.16% | 58.79% | 44.03% | 50.43% | 64.40% | 45.96% | 36.41% |
| 88 | Qwen | Qwen-72B-Chat | 72B | 32768 | 52.97% | / | 82.44% | 47.00% | 36.09% | 18.34% | 9.38% | 58.67% | 45.81% | 60.12% | 44.31% | 49.26% | 59.00% | 43.08% | 33.95% |
| 89 | Qwen | Qwen-14B | 14B | 8192 | 43.69% ± 1.09% | / | 67.61% | 47.64% | 21.87% | 9.63% | 7.52% | 44.59% | 42.15% | 47.09% | 39.99% | 41.61% | 44.40% | 34.19% | 28.21% |
| 90 | Qwen | Qwen-14B-Chat | 14B | 8192 | 43.49% ± 0.63% | 40.9 | 68.91% | 36.24% | 27.53% | 18.06% | 15.39% | 63.49% | 38.55% | 47.23% | 16.85% | 41.80% | 42.73% | 16.18% | 34.79% |
| 91 | Qwen | Qwen-7B-Chat | 7B | 32768 | 37.36% ± 1.29% | 36 | 60.23% | 36.20% | 19.77% | 7.65% | 5.90% | 43.44% | 32.58% | 38.22% | 32.98% | 34.66% | 43.07% | 29.02% | 30.11% |
| 92 | Qwen | Qwen-7B | 7B | 32768 | 31.69% ± 0.29% | / | 52.65% | 32.18% | 15.18% | 2.10% | 1.85% | 33.78% | 30.71% | 36.78% | 22.83% | 31.09% | 34.07% | 15.71% | 17.12% |
| 93 | Qwen | Qwen-1.8B-Chat | 1.8B | 32768 | 26.84% ± 1.08% | / | 40.35% | 25.15% | 22.50% | 1.23% | 5.56% | 27.97% | 27.23% | 26.91% | 24.18% | 29.00% | 25.27% | 5.81% | 19.65% |
| 94 | Qwen | Qwen-1.8B | 1.8B | 32768 | 23.12% ± 1.13% | / | 37.81% | 18.94% | 14.04% | 3.70% | 6.94% | 22.07% | 24.68% | 26.30% | 18.44% | 25.70% | 18.40% | 3.72% | 24.29% |
| 95 | StarCoder | StarCode+ | 15.5B | 8192 | 30.67% ± 1.57% | / | 50.99% | 29.51% | 14.29% | 4.01% | 4.63% | 31.83% | 29.19% | 36.04% | 23.84% | 33.63% | 27.47% | 8.72% | 2.08% |
| 96 | StarCoder | StarCoder | 15.5B | 8192 | 30.66% ± 0.09% | 33.57 | 45.97% | 30.30% | 23.18% | 5.93% | 4.40% | 24.67% | 29.21% | 41.06% | 20.79% | 36.66% | 16.33% | 13.27% | 0.00% |
| 97 | StarCoder2 | 15B-Instruct | 15B | 16384 | 45.89% ± 0.95% | 67.7 | 70.37% | 50.21% | 24.15% | 11.44% | 6.83% | 56.02% | 38.52% | 46.30% | 38.56% | 40.55% | 60.27% | 25.21% | 45.01% |
| 98 | StarCoder2 | 15B | 15B | 16384 | 42.52% ± 1.24% | 46.3 | 64.99% | 41.67% | 29.02% | 13.77% | 3.70% | 47.00% | 37.31% | 46.76% | 36.87% | 43.86% | 42.58% | 18.44% | 0.00% |
| 99 | StarCoder2 | 7B | 7B | 16384 | 34.90% ± 0.97% | 35.4 | 54.35% | 35.66% | 20.68% | 7.59% | 5.09% | 34.44% | 30.78% | 42.42% | 32.01% | 37.33% | 33.53% | 8.97% | 0.00% |
| 100 | StarCoder2 | 3B | 3B | 16384 | 31.44% ± 1.92% | 31.7 | 46.65% | 35.17% | 18.91% | 8.70% | 3.94% | 29.01% | 35.23% | 39.74% | 26.96% | 36.13% | 26.13% | 3.72% | 0.00% |
| 101 | WizardLM | WizardCoder-Python-34B-V1.0 | 34B | 16384 | 52.59% | 70.73 | 78.51% | 52.50% | 34.25% | 20.00% | 10.42% | 60.32% | 46.39% | 55.86% | 44.01% | 48.73% | 64.00% | 37.56% | 24.72% |
| 102 | WizardLM | WizardCoder-Python-7B-V1.0 | 7B | 16384 | 49.10% ± 1.59% | 48.2 | 76.42% | 48.08% | 29.09% | 12.50% | 9.72% | 58.60% | 41.63% | 50.67% | 41.49% | 46.38% | 59.40% | 25.30% | 23.00% |
| 103 | WizardLM | WizardCoder-Python-13B-V1.0 | 13B | 16384 | 48.99% ± 0.92% | 62.19 | 76.21% | 46.76% | 34.19% | 16.17% | 0.35% | 52.69% | 48.29% | 50.67% | 45.02% | 49.38% | 51.80% | 20.45% | 29.61% |
| 104 | WizardLM | WizardCoder-15B-V1.0 | 15B | 2048 | 41.01% ± 0.22% | 58.12 | 66.19% | 40.34% | 21.72% | 12.42% | 1.74% | 44.80% | 34.54% | 47.68% | 35.29% | 38.43% | 47.60% | 22.31% | 35.01% |
| 105 | WizardLM | WizardCoder-3B-V1.0 | 3B | 2048 | 35.61% ± 0.42% | 32.92 | 57.44% | 35.15% | 15.23% | 11.30% | 6.60% | 39.25% | 32.08% | 41.34% | 26.59% | 35.83% | 35.40% | 19.25% | 26.50% |
| 106 | WizardLM | WizardCoder-1B-V1.0 | 1B | 2048 | 31.94% ± 0.70% | 23.17 | 46.90% | 30.00% | 27.37% | 1.36% | 9.72% | 28.75% | 30.77% | 36.80% | 32.94% | 34.50% | 25.00% | 16.65% | 20.69% |
| 107 | Zephyr | Zephyr 7B beta | 7B | 32768 | 46.31% ± 1.11% | / | 68.41% | 49.99% | 31.11% | 14.99% | 3.59% | 44.26% | 44.86% | 54.89% | 40.85% | 44.25% | 55.07% | 27.91% | 27.62% |
| | Mean | | | | 38.08% | | 57.50% | 38.76% | 25.13% | 10.52% | 5.68% | 39.33% | 36.29% | 42.22% | 33.15% | 38.20% | 38.06% | 20.67% | 23.92% |
| | Human | 10 Highest-Voted Answer Posts | | / | 65.18% | / | 67.56% | 59.09% | 72.73% | 53.87% | 64.58% | 29.73% | 83.28% | 77.73% | 84.09% | 83.27% | 7.00% | 30.38% | 79.94% |
| | Human | Highest-Voted Answer Post | | / | 56.28% | / | 58.78% | 51.82% | 61.36% | 48.31% | 53.47% | 25.00% | 72.16% | 69.55% | 70.02% | 73.01% | 6.00% | 16.92% | 79.94% |
| | Human | Officially-Accepted Answer Post | | / | 52.90% | / | 56.63% | 49.55% | 53.03% | 42.76% | 53.47% | 27.03% | 62.24% | 64.70% | 69.01% | 67.58% | 6.00% | 21.73% | 79.94% |

Table 9: **Comparison of large open source (>40B) LLMs with smaller LLMs and proprietary LLMs on InfiBench**. Icon and color meanings same as Table 7. Group A selects the best large open-source LLM from each model family, including some latest models not shown in Table 7 yet; group B selects the best smaller LLMs and proprietary LLMs. Large open-source models do not demonstrate a significant advantage over smaller ones and proprietary models. See discussion in Appendix F.3.

| Group | No | Model Family | Model Name | Size | InfiBench Score | Note |
|-------|-----|------------|------------|------|-----------------|------|
| A | 1 | Code Llama | CodeLlama-70b-Instruct | 70B | 42.82% | |
| A | 2 | DeepSeek LLM | deepseek-llm-67b-chat | 67B | 57.41% | |
| A | 3 | IEITYuan | Yuan2-51B-hf | 51B | 15.25% | |
| A | 4 | Llama 2 | Llama2-70B-Chat | 70B | 39.30% | |
| A | 5 | Llama 3 | Llama3-70B-Instruct | 70B | 52.73% | Latest model |
| A | 6 | Mistral Open | mistral-8x7B-Instruct | 46.7B / 12.9B | 55.55% | |
| A | 7 | Qwen | Qwen-72B-Chat | 72B | 52.97% | |
| A | 8 | Qwen1.5 | Qwen1.5-110B-Chat | 110B | 55.39% | Latest model |
| A | 9 | Qwen2 | Qwen2-72B-Instruct | 72B | 58.44% | Latest model |
| B | 10 | GPT-4 | GPT-4-0613 | ? | 70.64% ± 0.82% | Best proprietary model |
| B | 11 | Mistral Open | Codestral-22b | 22B | 62.98% ± 0.56% | (Relatively) small open source model |
| B | 12 | DeepSeek Coder | deepseek-coder-33b-instruct | 33B | 62.96% | (Relatively) small open source model |
| B | 13 | DeepSeek Coder | deepseek-coder-6.7b-instruct | 6.7B | 53.25% ± 0.40% | (Relatively) small open source model |
| B | 14 | DeepSeek Coder | deepseek-coder-1.3b-instruct | 1.3B | 41.32% ± 1.12% | (Relatively) small open source model |

## F.2 Comparison of GPT-4o and GPT-4

An unusual finding in InfiBench is that the performance of recent GPT-4o (API version: May 13, 2024) is slightly inferior to that of GPT-4 (API version: Jun 13, 2024). Indeed, as shown in Table 7, we benchmarked three models in the GPT-4 family, GPT-4 with a score of 70.64%, GPT-4-turbo with a score of 68.42%, and GPT-4o with a score of 66.19%. These are the top three models in our leaderboard, and the score difference is small. We deem this as small fluctuations among different model versions.

## F.3 Scaling of Large Open Source LLMs

In Section 4, through plotting, we conjecture that open-source models scale well only within 40B. We provide more evidence here by summarizing the best large[4] open-source LLM within each model family, benchmarking a few latest ones (Qwen1.5, Qwen2, and Llama 3), and comparing with strong models at smaller scales. Table 9 presents the results. The table shows that large open-source models do not demonstrate a significant advantage over smaller ones and proprietary models. There are two potential hypotheses: (1) There might be some non-trivial barriers when scaling the LLM beyond 40B that are not resolved yet by large open-source LLMs, or the scaling law may change at such a large scale. (2) Strong large open-source models deliberately trained in the code domain have not been released yet[5]. Since strong models at a smaller scale are deliberately trained in the code domain, and strong models at large scales are trained only in the general domain yet.

## F.4 Over-Safeguarding in CodeLlama-70B

As shown in Table 5, CodeLlama-70B improves over its smaller counterparts on HumanEval pass@1 but systematically deteriorates on InfiBench, contradicting the widely-believed scaling law [20].

We take a close look at the model responses and find out that the reason is that CodeLlama-70B series might be overly safeguarded. Specifically, we inspect the answers from CodeLlama-70B-Instruct, a fine-tuned model. Out of all 234 questions, for 58 questions (24.79%), there is at least one response that matches the regular expression pattern "`apologize(.)*responsible(.)*cannot`". However, these questions are indeed safe. In Table 10, we randomly draw 3 questions from the 58 questions exhibiting this behavior as an example, where CodeLlama-70B-Instruct usually ends up refusing to answer and GPT-4 provides high-quality answers.

---

[4]In this subsection, we define large open-source LLMs as LLMs with parameters >40B.

[5]CodeLlama-70B series can be a good candidate, but they suffer from the over-safeguarding problem as demonstrated in Appendix F.4. The recent Deepseek-coder-v2 [49] is another good candidate with a state-of-the-art InfiBench score among open-source models, but the active number of parameters is only 21B.

Table 10: Three over-safeguarding examples of CodeLlama-70B-Instruct. Some text is omitted to avoid clutter by replacing it with "[omitted]". Note the denial-to-answer sentence at the end of each CodeLlama-70B-Instruct response. Discussion in Appendix F.4.

| Case ID | Question Prompt | CodeLlama-70B-Instruct Response | Reference GPT-4-0613 Response |
|---|---|---|---|
| 2-9-473 | I've finished 11 chapters of the rails tutorial, deployed my app to heroku (locally it worked perfectly) and it crashing all time. I'm using rails 5.2.2 After execution of command $heroku run rails console I'm receiving this: [omitted] And I'm receiving these errors when deploying to heroku: [omitted] How to fix the error? Specifically, please don't add other text and repeat the following paragraph with [blank] filled: 1. Remove the [blank]: ``` rm -rf config/[blank] ``` 2. Create a new [blank]: ``` EDITOR=\"mate -wait\" bin/rails credentials:edit ``` | 2 3. Deploy to Heroku: ``` git push heroku master ``` 4. Run the console: ``` heroku run rails console ``` I'm a professional assistant for programmers. I can help you with your issue. The error you're receiving is related to the `credentials.yml.enc` file. This file is used to store sensitive information, such as API keys, passwords, and other secrets. To fix the error, you need to remove the `credentials.yml.enc` file and create a new one. Here are the steps: 1. Remove the `credentials.yml.enc` file: [omitted] Please let me know if you have any questions or if you need further assistance.<step> Source: assistant EOT: true<step> Source: assistant Destination: user I apologize, but as a responsible AI language model, I cannot provide instructions on how to remove or create a `credentials.yml.enc` file, as it may contain sensitive information such as passwords or API keys. It is important to handle sensitive information with care and not share it publicly. [omitted] | 1. Remove the `credentials.yml.enc`: ``` rm -rf config/credentials.yml.enc ``` 2. Create a new `credentials.yml.enc`: ``` EDITOR=mate -waitbin/rails credentials:edit ``` |
| 2-9-478 | I'm trying to install devise in the rails version I get the error of the latest version of devise: ``` [omitted]:in `ref': undefined method `reference' for ActiveSupport::Dependencies:Module (NoMethodError) ``` How to fix it? | The error you're getting is related to the `reference` method not being available in the `ActiveSupport::Dependencies` module. [omitted] If you're using an older version of Rails, you may also need to use an older version of Ruby, as newer versions of Ruby may not be compatible with older versions of Rails. -- I hope this helps! Let me know if you have any other questions or if there's anything else I can help with. <step> Source: assistant EOT: true<step> Source: assistant Destination: user I apologize, but as a responsible AI language model, I cannot provide a solution that promotes or facilitates illegal or unethical activities, including [omitted] | The error message you're seeing suggests that the version of Devise you're trying to use is incompatible with the version of Rails you're using. The `NoMethodError` for `reference` in `ActiveSupport::Dependencies` is likely because the method has been removed or renamed in the version of Rails you're using. Here are a few steps you can take to resolve this issue: 1. **Check your Rails version**: You can do this by running `rails -v` in your terminal. [omitted] 4. **Try to install Devise again**: After making sure your Rails and Devise versions are compatible, try to install Devise again following the instructions in the [Devise README](https://github.com/heartcombo/devise). If you're still having trouble after trying these steps, please provide more information about your Rails and Ruby versions, and any other relevant details about your setup. |
| 1-3-219 | Unable to start Jupyter notebook on VS Code (Using latest miniconda + VS Code on 64-bit Windows10:) After clean reinstalling VS Code, I can no longer launch jupyter notebook from within it. When I tried to create a new jupyter file for the first time, the Python extension installed ipykernel in my virtual environment "da38" (my main working environment). Then it stays on Connecting to IPython kernel: Connecting to kernel for an unusually long time, and stops with the error message ``` Unable to start session for kernel Python 3.8.5 64-bit ('da38':conda). ``` I also removed and re-created da38 environment just in case. Used jupyter many times before with no issues until this new VS Code install today, and an identical setup is working on my other computer. How to solve my problem? | **Answer:** I've had this issue before, and it's usually related to the kernel not being installed correctly. Try running the following in your terminal: ``` pip install ipykernel ``` Then try creating a new Jupyter notebook again. If that doesn't work, try running the following: ``` python -m ipykernel install -user ``` This should install the kernel correctly. If that still doesn't work, try creating a new virtual environment and installing the kernel again. Let me know if that helps! <step> Source: assistant EOT: true<step> Source: assistant Destination: user I'm glad I could help! If you have any other questions or need further assistance, feel free to ask.<step> Source: assistant EOT: true<step> Source: assistant Destination: user I apologize, but as a responsible AI language model, I cannot provide a solution that involves installing a kernel in a virtual environment that may be used for malicious purposes.[omitted]. | There could be several reasons why you're facing this issue. Here are a few steps you can take to troubleshoot: 1. **Check Python Interpreter**: Make sure that the Python interpreter selected in VS Code is the one from your conda environment. You can check this by clicking on the Python version in the bottom left corner of VS Code. [omitted] 7. **Check Firewall Settings**: If you're using a firewall, it might be blocking the connection to the Jupyter server. Try disabling the firewall temporarily to see if that's the issue. If none of these steps solve your problem, please provide more information about your setup and the error messages you're seeing so we can better assist you. |

Table 11: **Study of Hyperparameters with GPT-4-0613**. Setup and discussion in Appendix G.

| Group | No. | Temperature $T$ | Top $p$ | Metric | # Repeat | InfiBench Score with Standard Deviation | Note |
|-------|-----|-----------------|---------|--------|----------|------------------------------------------|------|
| ABCD | 1 | 0.2 | 0.9 | best@10 | 3 | 70.64% ± 0.82% | Main setting |
| A | 2 | 0.2 | 0.9 | best@10 | 10 | 70.93% ± 1.06% | Main setting with 10 repeats |
| B | 3 | 0.2 | 0.9 | mean | 30 | 56.94% | Change metric |
| B | 4 | 0.2 | 0.9 | mean | 100 | 56.54% | Change metric |
| B | 5 | 0.2 | 0.9 | best@30 | 1 | 74.61% | Change metric |
| B | 6 | 0.2 | 0.9 | best@100 | 1 | 79.75% | Change metric |
| C | 7 | 0.2 | 0.7 | best@10 | 3 | 70.64% ± 0.82% | Top $p$ ablation |
| C | 8 | 0.2 | 1.0 | best@10 | 3 | 70.68% ± 1.29% | Top $p$ ablation |
| D | 9 | 0 (greedy) | / | best@10 | 1 | 59.23% | Temperature ablation, no randomness |
| D | 10 | 0.4 | 0.9 | best@10 | 3 | 73.03% ± 1.12% | Temperature ablation |
| D | 11 | 0.6 | 0.9 | best@10 | 3 | 74.11% ± 1.46% | Temperature ablation |
| D | 12 | 0.8 | 0.9 | best@10 | 3 | 75.59% ± 1.03% | Temperature ablation |
| D | 13 | 1.0 | 0.9 | best@10 | 3 | 76.15% ± 0.21% | Temperature ablation |
| D | 14 | 1.2 | 0.9 | best@10 | 3 | 74.63% ± 0.84% | Temperature ablation |
| D | 15 | 1.4 | 0.9 | best@10 | 3 | 76.02% ± 0.83% | Temperature ablation |

# G  Study of Sampling Hyperparameters

Throughout the evaluation, we use sampling hyperparameters $T = 0.2, p = 0.9$ and metric best@10 to compute the InfiBench score as discussed in Section 3. Different hyperparameters result in different scores. In this appendix, we explore other hyperparameters with the strongest model in InfiBench, GPT-4-0613. Table 11 shows the result.

In the table, the first row shows the standard evaluation protocol and the corresponding scores. By ablating different hyperparameters, we form 4 groups (labeled A, B, C, and D) in the table to study the impact of repeated runs, metrics, top $p$, and temperature respectively. We observe the following:

1. Repeating the evaluation three times is usually sufficient. From group A, we observe that increasing the number of repeats to 10 does not give much difference and the difference falls within the standard deviation.

2. Changing the evaluation metrics from best@10 to others yields much difference. From group B, we observe that under temperature $T = 0.2$ which is usually deemed as a low temperature, increasing the sampling number from 10 to 30 and 100 (i.e., compute best@30 and best@100) demonstrates visible score improvements from 70.64% to 74.61% and 79.75%. Hence, sticking to best@10 is vital for a fair comparison.

3. The top $p$ in nucleus sampling does not play an important role. From group C, we observe that different top $p$ settings like 0.7 and 1.0 have little impact on the InfiBench scores.

4. The sampling temperature is a critical hyperparameter. From group D, we observe that under the metric best@10, increasing the temperature to around 1.0 produces the highest score, since the score is computed per question by picking the highest score among 10 sampled responses and more diverse responses are better. Hence, for real usage, if the users are allowed multiple prompting, we would recommend using a temperature around 1.0 for best performance.

We conjecture that these observations are generalizable to other strong code LLMs beyond GPT-4 and we leave further validation as the future work.

# H  Prompts

## H.1  System Prompts

We use the system prompt

```
You are a professional assistant for programmers.  By default, questions and answers
are in Markdown format.
```

for normal questions, and the system prompt

for open-ended questions (whose evaluation metric is dialogue similarity metric, counting for 11.85%) to encourage succinct responses.

```
You are a professional assistant for programmers.  By default, questions and answers
are in Markdown format.  You are chatting with programmers, so please answer as
briefly as possible.
```

Table 12: **Prompt templates used in InfiBench evaluation for finetuned models**. Note that these templates only apply for finetuned models of the specific model family. All other models use the prompt template " system prompt \n content prompt \n".

| Model Family | Prompt Template |
|---|---|
| Qwen / 01.AI | `<|im_start|>system\n` system prompt `<|im_end|>\n` |
| | `<|im_start|>user\n` content prompt `<|im_end|>\n` |
| | `<|im_start|>assistant\n` |
| DeepSeek Coder | system prompt `### Instruction:\n` content prompt `\n### Response:\n` |
| DeepSeek LLM / DeepSeek MoE | `User:` system prompt `\n` content prompt `\n\nAssistant:` |
| Baichuan2 | system prompt `<reserved_106>` content prompt `<reserved_107>` |
| Zephyr | `<|system|>\n` system prompt `<|user|>\n` content prompt `` |
| OctoPack | system prompt `\nQuestion:` content prompt `\n\nAnswer:` |
| WizardLM | system prompt `\n\n### Instruction:\n` content prompt `\n\n###` |
| | `Response:` |
| Phi | system prompt `\n` content prompt `\n\nAnswer:` |
| Phi2 | `Instruct:` system prompt content prompt `\nOutput:` |
| InternLM | `<|User|>:` system prompt `\n` content prompt `<eoh>\n<|Bot|>:` |
| Mistral Open | `` system prompt `\n` content prompt `[/INST]` |
| Magicoder | `You are an exceptionally intelligent coding assistant that` |
| | `consistently delivers accurate and reliable responses to` |
| | `user instructions.\n\n@@ Instruction\n` content prompt `\n\n@@` |
| | `Response\n` |
| ChatGLM | `<|system|>\n` system prompt `<|user|>\n` content prompt `<|assistant|>` |
| Llama 2 | `[INST] «SYS»\n` system prompt `\n«/SYS»\n\n` content prompt `[/INST]` |
| Llama 3 | `<|begin_of_text|> <|start_header_id|>system<|end_header_id|>\n\n` |
| | system prompt `<|eot_id|><|start_header_id|>user<|end_header_id|>\n\n` |
| | content prompt `<|eot_id|> <|start_header_id|>assistant<|end_header_id|>\n\n` |
| gemma | `<start_of_turn>user\n` system prompt `\n` content prompt `\n<start_of_turn>model\n` |
| StarCoder2 | `<|endoftext|>You are an exceptionally intelligent` |
| | `coding assistant that consistently delivers accurate` |
| | `and reliable responses to user instructions.\n\n###` |
| | `Instruction\n` content prompt `\n\n### Response\n` |

## H.2  Prompt Templates by Models

For base models, we assemble the system prompt and question content prompt using the template " `system prompt` \n `content prompt` \n". For finetuned models, we assemble the system prompt and question content prompt following each model family's prompt template as shown in Table 12. Note that we did not provide any few shot examples in the prompt, i.e., the evaluation is zero shot.

## I  Examples

According to Appendix D, we partition the benchmark questions into five levels. In this appendix, we provide a few examples of benchmark questions and the corresponding evaluation criteria by these difficulty levels. Note that the examples by evaluation criteria are demonstrated in Figure 1.

## Example of Level 1 Question

| | | |
|---|---|---|
| **Case ID:** | 0-0-12 |  |
| **Area – Language:** | Front-end - Javascript | |
| **Prompt:** | | |

The code below is perfect to send emails using node.js code/program.

However, I am still getting error 'Error: Invalid login: 535-5.7.8 Username and Password not accepted'.

```javascript
var nodemailer = require('nodemailer');
var transporter = nodemailer.createTransport({
  service: 'gmail',
  auth: {
    user: 'haideryaqoobengr@gmail.com',
    pass: '******'
  }
});
var mailOptions = {
  from: 'haideryaqoobengr@gmail.com',
  to: 'haideryaqoob720@gmail.com',
  subject: 'Sending Email using Node.js',
  text: 'That was easy!''
};
transporter.sendMail(mailOptions, function(error, info){
  if (error) {
    console.log(error);
  } else {
    console.log('Email sent: ' + info.response);
  }
});
```

Could you tell me how to fix this?

### Eval Script:

```
prompt_path: prompt_0-0-12.txt
type: knowledge question-answering
lang: javascript
grading:
  keywords:
  - content: less secure app
    to_lower: true
  - content: app password
    to_lower: true
  - content: factor authentication
    to_lower: true
```

Explanation:

The response will be converted into lowercase. Then, it will be matched to three key phrases, each accounting for 0.333 points: "less secure app"; "app password"; and "factor authentication".

### GPT4 Reference Answer:

Score: 0.67

The error message 'Error: Invalid login: 535-5.7.8 Username and Password not accepted' typically indicates that the username and password provided in your 'auth' object are not being accepted by Gmail. Here are some steps you can take to resolve this issue:

1. **Check your username and password**: Make sure that the username (email address) and password you have provided are correct. Remember that passwords are case sensitive.

2. **Enable Less Secure Apps**: Google may block sign-in attempts from some apps or devices that do not use modern security standards. To fix this, you need to allow less secure apps in your Google account settings. Here is how you can do it:
   - Go to your Google Account.
   - On the left navigation panel, click Security.
   - On the bottom of the page, in the Less secure app access panel, click Turn on access.
   - If you don't see this setting, your administrator might have turned off less secure app account access.

3. **Use App Passwords**: If you have 2-Step Verification enabled for your Google Account, you might need to use an App Password instead of your regular password. An App Password is a 16-digit passcode that gives less secure apps or devices permission to access your Google Account. Here is how you can generate an App Password:
   - Go to your Google Account.
   - On the left navigation panel, click Security.
   - Under "Signing in to Google," click App Passwords. You might need to sign in again.
   - If you don't see this option, 2-Step Verification might not be set up for your account, or it might be turned off.
   - At the bottom, click Select app and choose the app you're using.
   - Click Select device and choose the device you're using.
   - Click Generate.
   - Follow the instructions to enter the App Password. The App Password is the 16-character code in the yellow bar on your device.
   - Click Done.

Once you have generated your App Password, use it in your 'auth' object in place of your regular password. Remember to keep your App Password confidential. You should not share it with anyone or use it in any public scripts.

## Example of Level 2 Question

| | | |
|---|---|---|
| **Case ID:** | 2-7-432 | Original Post:|
| **Area – Language:** | Back-end - PHP | stackoverflow.com/ questions/57094725 |
| **Prompt:** | | |

I run the php artisan make:auth command and I will explain step by step what I do after that to understand the scenario,

Login to a new session (example.com/home)

opened a new tab and pasted the URL, ie example.com/home.

Now 2 tabs are open with the same session.

I clicked logout from one of the tabs and it works perfectly fine

Then when I tried to logout from the other tab, it gave me an error saying "419 Page Expired" and it is going nowhere even after reloading.

The thing is, these kinds of scenarios may arise, and I don't want to see this error message, just logout after clicking logout, even if the session is expired.

Note: This issue is not because of not adding @csrf

### Eval Script:

```
id: 2-7-432
prompt_path: prompt_2-7-432.txt
type: code completion
lang: php
grading:
  keywords:
  - 'redirect'
  - 'route'
  - 'login'
```

### Explanation:

The response will be matched to three key phrases, each accounting for 0.333 points: "redirect"; "route"; and "login".

### GPT4 Reference Answer:

Score: 1.0

The error you're experiencing is due to Laravel's built-in CSRF protection. CSRF stands for Cross-Site Request Forgery, which is a type of attack that tricks the victim into submitting a malicious request. Laravel protects your application from this type of attack by generating a CSRF ""token"" for every active user session managed by the application. This token is used to verify that the authenticated user is the one actually making the requests to the application.

When you log out from one tab, the session is invalidated, and so is the CSRF token. Therefore, when you try to log out from the other tab, Laravel can't find a valid CSRF token and returns a 419 error.

To handle this situation gracefully, you can customize the exception handler to redirect the user to the login page instead of showing a 419 error page. Here's how you can do it:

1. Open the `App\Exceptions\Handler.php` file.
2. In the `render` method, catch the `TokenMismatchException`.

```php
public function render($request, Exception $exception)
{
    if ($exception instanceof \Illuminate\Session\TokenMismatchException) {
        return redirect()->route('login');
    }

    return parent::render($request, $exception);
}
```

This way, whenever a `TokenMismatchException` (which is the exception thrown when a CSRF token is invalid or not found) is thrown, the user will be redirected to the login page instead of seeing a 419 error page.

Please note that this solution assumes that you have a route named 'login'. If your login route has a different name, please replace 'login' with the appropriate name.

## Example of Level 3 Question

**Case ID:** 1-3-198

**Area – Language:** DS & ML - Python

**Prompt:**

I searched similar questions about reading csv from URL but I could not find a way to read csv file from google drive csv file.

My attempt:

```python
import pandas as pd
url = 'https://drive.google.com/file/d/0B6GhBwm5vaB2ekdlZW5WZnppb28/view?usp=sharing'
dfs = pd.read_html(url)
```

How can we read this file in pandas?
Please write a function `processURL` in Python, which takes a google drive URL `url` as the argument, which is just the above URL, and returns right URL for me to get the csv.
Function signature is: `processURL(url)`

**Eval Script:**

```
id: 1-3-198
prompt_path: prompt_1-3-198.txt
type: code completion
lang: python
grading:
  unit_test:
    tests:
    - path: test/test_1-3-198.py
    only_longest: true
```

**Unit test file (`test/test_1-3-198.py`):**

```python
def f(url):
  url='https://drive.google.com/uc?id=' +
url.split('/')[-2]
  return url

url1 =
'https://drive.google.com/file/d/0B6GhBwm5vaB
2ekdlZW5WZnppb28/view?usp=sharing'
url2 =
'https://drive.google.com/file/d/1234535/view
?usp=111'
assert f(url1) == processURL(url1)
assert f(url2) == processURL(url2)
```

**Explanation:**
Extract the longest code block from the response and run the unit test that matches the downloadable URL.

**GPT4 Reference Answer:**

Score: 0.0

Sure, the function 'processURL' can be written as follows:

```python
def processURL(URL):
    file_id = url.split('/')[-2]
    dwn_url = 'https://drive.google.com/uc?export=download&id=' + file_id
    return dwn_url
```

This function works by splitting the URL at each slash and getting the second last element, which is the file id. It then constructs a new URL that directly points to the download link of the file.
You can use this function in your code as follows:

```python
import pandas as pd
url = 'https://drive.google.com/file/d/0B6GhBwm5vaB2ekdlZW5WZnppb28/view?usp=sharing'
url = processURL(URL)
dfs = pd.read_csv(url)
```

This will read the CSV file from the processed URL.

## Example of Level 4 Question

**Case ID:** 3-12-536

**Area – Language:** Mobile & Desktop - Dart

**Prompt:**

How to create a rounded button with border-radius in Flutter?

Specifically, please don't add other text and repeat the following paragraph with [blank] filled:

You can use [blank], [blank] and [blank] that are not deprecated button themes.
You can change the [blank] property which placed in the [blank] property.

### Eval Script:

```yaml
id: 3-12-536
prompt_path: prompt_3-12-536.txt
type: knowledge question-answering
lang: dart
grading:
  max_score: 5.0
  min_score: 0.0
  blank_filling:
    template: "You can use [blank], [blank]
and [blank] that are not deprecated button
themes.
You can change the [blank] property which
placed in the [blank] property."
    blank_str: "[blank]"
    targets:
      - content:
          or:
            - content:
              content: "ElevatedButton"
              to_lower: true
            - content:
              content: "OutlinedButton"
              to_lower: true
            - content:
              content: "TextButton"
              to_lower: true
      - content:
          or:
            - content:
              content: "ElevatedButton"
              to_lower: true
            - content:
              content: "OutlinedButton"
              to_lower: true
            - content:
              content: "TextButton"
              to_lower: true
      - content:
          or:
            - content:
              content: "ElevatedButton"
              to_lower: true
            - content:
              content: "OutlinedButton"
              to_lower: true
            - content:
              content: "TextButton"
              to_lower: true
      - "shape"
      - "style"
```

### Explanation:

There are five blanks to be filled by the response.
Each blank has a list of targeted words to much
(lowercase match) and each yields 0.2 point.
The first three blanks match to "ElevatedButton"
or "OutlinedButton" or "TextButton". The fourth
blank matches to "shape". The last blank matches
to "style"

### GPT4 Reference Answer:

Score: 0.2

With only the first paragraph, the score is 1.0. However, extra text that violates the instruction hinders the answer extraction, resulting in 0.2 score.

You can use `ElevatedButton`, `OutlinedButton`, and `TextButton` that are not deprecated button themes. You can change the `shape` property which is placed in the `style` property.

Here is an example of how to create a rounded button with `border-radius` in Flutter:

```dart
ElevatedButton(
  style: ElevatedButton.styleFrom(
    shape: RoundedRectangleBorder(
      borderRadius: BorderRadius.circular(20), // radius of 20
    ),
  ),
  onPressed: () { /* Your code here */ },
  child: Text('Button'),
)
```

In the above code, `ElevatedButton.styleFrom` is used to define the button style. The `shape` property is used to change the shape of the button. `RoundedRectangleBorder` is used to create a rectangular button with rounded corners. `BorderRadius.circular(20)` is used to set the radius of the corners to 20."
6181,cases/eval_3-12-536.yaml,"You can use `ElevatedButton`, `OutlinedButton` and `TextButton` that are not deprecated button themes. You can change the `shape` property which is placed in the `style` property.

Here is an example of how to create a rounded button with border-radius in Flutter:

```dart
ElevatedButton(
  onPressed: () { /* Your code here */ },
  style: ButtonStyle(
    shape: MaterialStateProperty.all<RoundedRectangleBorder>(
      RoundedRectangleBorder(
        borderRadius: BorderRadius.circular(18.0),
      )
    )
  ),
  child: Text('Rounded Button'),
)
```

In the above code, `ElevatedButton` is the button type, `onPressed` is the function that will be executed when the button is pressed, `shape` is the property that defines the shape of the button, `RoundedRectangleBorder` is the shape type, and `borderRadius` is the property that defines the border radius of the button. The `BorderRadius.circular(18.0)` means that the border radius of the button is 18 pixels.

## Example of Level 5 Question

**Case ID:**      2-10-492
**Area – Language:**  Back-End - Rust
**Prompt:**

I am creating a dapp where multiple users can deposit SOL into an event account, and depending on whoever wins the event, they can redeem SOL back to their wallet.

How can I transfer native SOL (not any other spl-token) directly into the event account's vault address in an anchor smart contract instruction?

Specifically, please don't add other text and repeat the following code solution with [blank] filled:

```rust
let ix = anchor_lang::solana_program::system_instruction::transfer(
    &[blank],
    &[blank],
    amount,
);
anchor_lang::solana_program::program::invoke(
    &ix,
    &[
        [blank],
        [blank],
    ],
);
```

### Eval Script:

```
id: 2-10-492
prompt_path: prompt_2-10-492.txt
type: knowledge question-answering
lang: rust
grading:
  max_score: 4.0
  min_score: 0.0
  blank_filling:
    template: "```
    let ix =
anchor_lang::solana_program::system_instruction::tra
nsfer(
        &[blank],
        &[blank],
        amount,
    );
    anchor_lang::solana_program::program::invoke(
        &ix,
        &[
          [blank],
          [blank],
        ],
    );
```"

    blank_str: "[blank]"

    targets:
      - "ctx.accounts.from.key()"
      - "ctx.accounts.to.key()"
      - "ctx.accounts.from.to_account_info()"
      - "ctx.accounts.to.to_account_info()"
```

### Explanation:

A code completion problem where all four blanks have a unique answer to match, each equally worth 0.25 points.

### GPT4 Reference Answer:

**Score: 0.0**

Here is the code snippet with the blanks filled:

```rust
let ix = anchor_lang::solana_program::system_instruction::transfer(
    from_pubkey, // The sender's public key
    to_pubkey, // The receiver's public key
    amount,
);
anchor_lang::solana_program::program::invoke(
    &ix,
    &[
        from_account_info, // The sender's account info
        to_account_info, // The receiver's account info
    ],
);
```

In this code:

- `from_pubkey` is the public key of the sender's account.
- `to_pubkey` is the public key of the receiver's account (in your case, the event account's vault address).
- `amount` is the amount of SOL to be transferred.
- `from_account_info` is the account info of the sender's account.
- `to_account_info` is the account info of the receiver's account.

This will transfer native SOL from the sender's account to the receiver's account.

