# OpenReview forum: "InfiBench: Evaluating the Question-Answering Capabilities of Code Large Language Models"
_NeurIPS.cc/2024/Datasets_and_Benchmarks_Track — NeurIPS 2024 Track Datasets and Benchmarks Poster_

### Official Review · Reviewer_bRXT · 2024-07-08
**A well-curated and in-time dataset for evaluating code QA**

**Rating:** 8
**Confidence:** 4
**Correctness:** Dataset construction and evaluation a…
**Clarity:** The paper is clearly written and easy…

**Review:**

See below.

**Strengths:**

Evaluating LLMs' capability in QA for code is a critical yet challenging task. The proposed InfiBench is a timely benchmark that will facilitate the development of LLM-based code assistants through the chat interface.
* The dataset covers diverse topics with various evaluation metrics.
* Human annotation, particularly the paraphrasing stage, ensures prompt compatibility with popular LMs and reduces memorization effect.
* Existing and ongoing implementation works make the benchmark easily accessible.
* Extensive evaluation demonstrates the turning point in scaling law among other interesting comparisons between model families, such as pretrained / finetuned, or general / code-specific.

**Additional Feedback:**

* (L235) "For generic models" appears twice.
* (L243) "Table 6" should probably be "Table 4".
* (L252) "pertaining" >> "pretraining"

**Documentation:**

Yes

**Limitations:**

Yes

**Opportunities For Improvement:**

* While the dataset size is determined by referencing HumanEval and constrained by annotation budget, expanding it to larger scale (e.g. around 500) will help to measure small improvements (<1%) with good statistical significance.

**Relation To Prior Work:**

Yes

**Summary And Contributions:**

The paper introduces a new dataset for evaluating QA capability of LLMs in code domain. Questions are sourced from the public StackOverflow and then annotated by industry domain experts for selection, paraphrasing, and determining sample-specific evaluation criteria. Further calibration and post-filtering result in 234 questions in the final dataset, covering 4 question types and 4 evaluation metrics. Considering the complexity of evaluating open-ended QA, the dataset leverages both lexical matching and unit testing. Evaluation over extensive LLMs reveals the turning point of scaling law, as well as other differences between pretrained versus instruction-tuned models, or general versus code-specific models.

---

> ### Author Rebuttal · Authors · 2024-08-19
>
> We thank the reviewer for the constructive and insightful comment! We are glad the reviewer acknowledges our InfiBench benchmark is timely, extensive, comprehensive, sound, and clearly presented. We answer your concerns and feedback below.
>
> > Concern / Feedback 1: While the dataset size is determined by referencing HumanEval and constrained by annotation budget, expanding it to larger scale (e.g. around 500) will help to measure small improvements (<1%) with good statistical significance.
>
> We appreciate the reviewer's feedback on the dataset size. As the reviewer rightly points out, the size is primarily constrained by the annotation budget. It's worth noting that the recent [SWE-bench Verified](https://openai.com/index/introducing-swe-bench-verified/) from OpenAI, which has 500 samples, is at a similar scale despite involving a large group of annotators. We are also aiming for 500 samples in our next release. The publication of this submission will encourage community involvement in benchmark expansion, which could help us reach our target size sooner. We are also continuously expanding the benchmark, with [40 new questions](https://github.com/infi-coder/infibench-evaluation-harness/tree/expand_set), and models are being benchmarked on them. Meanwhile, we plan to incorporate other reviewers' suggestions and increase the sampling ratio for less frequent but important topics in our expansion plan.
>
> > Concern / Feedback 2: Suggested corrections in "additional feedback."
>
> We are grateful for the reviewer's corrections and apologize for the errors. We have rectified these in the next version of the manuscript.
>
> -----
>
> In addition, we would like to highlight the **top general rebuttal post**, where we present additional human evaluation to compare which aligns with human ranking better between InfiBench evaluation and model(GPT-4o)-based evaluation along with our recent expansion of the benchmark (new questions and new benchmarked models). We plan to add these results in the next version of the manuscript.
>
> Thanks again for your insightful review! If there is any unclear point or your concerns remain, please let us know; we are happy to address them.

---

> > ### Author Response · Authors · 2024-08-29
> > **A Kind Reminder of Paper Discussion Period**
> >
> > Dear Reviewer bRXT,
> >
> > Thanks again for your appreciation of our work! As the end of the discussion (Aug 31 AoE) is approaching, we really appreciate it if you can consider our responses to your questions. We are very thankful for your comments and suggestions that helped improve our paper, and we will incorporate them in our revision. We also posted new results in the top rebuttal post. Please let us know if you have any questions, and we are more than happy to address them. Thanks!

---

> > > ### Comment · Reviewer_bRXT · 2024-08-29
> > >
> > > Thanks for the response and additional results. I will keep my score.

---

### Official Review · Reviewer_KYrR · 2024-07-23

**Rating:** 6
**Confidence:** 4
**Correctness:** Yes
**Clarity:** Yes, the paper is well written.

**Review:**

Strengths:
- A large-scale coding-related QA benchmark, four automatic metrics, and evaluation of 100 recent code LLMs.
- The data curation process considers the coverage of domains, realistic problems, and data contamination.


Weaknesses:
- The difference between the existing works is not very clear. Although the author mentioned three points in the introduction: 1) "Benchmarks beyond code generation are relatively few", it's not a strong motivation since ArenaHard already contains a lot of coding-related QAs. 2) "Independent code benchmarks are relatively few"; similarly, this is not a strong motivation. 3) Existing code benchmarks are saturating; there are many ways to address this issue, such as constructing a hard humaneval. I admit the contribution of providing a high-quality coding-related benchmark, but I don't think the current narrative offers a strong motivation.
- Lacking comparison with human judgment. This work mentioned four different metrics and said the domain experts decide the assignment of these metrics for each question. It's better to compute the correlation between the model ranks of automatic metrics and the model ranks of human judgment. You can sample 50 or 100 questions to compute the correlation, which is common in benchmarking works. Otherwise, it's hard to assess the reliability of the proposed metrics.

**Strengths:**

See review.

**Additional Feedback:**

A minor issue is that Section 4 is not very insightful. Most findings summarise the experiment results rather than provide insights.

**Documentation:**

Yes

**Ethics:**

no concerns

**Limitations:**

Yes, it' discussed in the Appendix.

**Opportunities For Improvement:**

See review.

**Relation To Prior Work:**

Yes

**Summary And Contributions:**

This work presents a new coding-related QA benchmark to assess the diverse capabilities of code LLMs, InfiBench, featuring 234 high-quality Stack Overflow questions across 15 programming languages. This work employs four model-free automatic metrics, keyword, blank filling, test cases, and text similarity, to evaluate correctness, with criteria set by domain experts. It evaluated over 100 recent code LLMs, yielding insights for the evolution of code LLMs.

Contributions:
- A large-scale coding-related QA benchmark
- Performance evaluation of 100 recent code LLMs

---

> ### Author Rebuttal · Authors · 2024-08-19
>
> We thank the reviewer for the constructive and insightful comment! We are delighted that the reviewer appreciates the contribution of our benchmark! We answer your concerns and questions below.
>
> > Concern 1: The difference between the existing works is not very clear. I admit the contribution of providing a high-quality coding-related benchmark, but I don't think the current narrative offers a strong motivation.
>
> Thanks for the comment and your appreciation. Yes, as mentioned, our key motivations are three-fold: (1) Benchmarks beyond code generation are relatively few. (2) Independent code benchmarks are relatively few. (3) Existing code benchmarks are saturating. Among them, the core motivation is **the demand for a reliable evaluation that shows room for improvement for the code model's ability to answer free-form code-related questions**. For this motivation, potential solutions are inferior to InfiBench:
>
> - ArenaHard: Though ArenaHard contains coding-related QAs, these questions lack human-annotated evaluation criteria. As a result, ArenaHard relies on GPT-4-turbo for pairwise comparison as the evaluation. In contrast, InfiBench leverages human-annotated criteria for evaluation. Our human evaluation experiment (see response to Concern 2) shows that model-based evaluation is less aligned with humans than InfiBench evaluation.
> Moreover, pairwise comparison in ArenaHard provides only a relative evaluation - it could not reflect what degree a question is fully answered and could not show room for improvement for top-performing models. In contrast, InfiBench reports an absolute score and fulfills the goals as mentioned earlier.
>
> - Constructing a hard humaneval: Though a hard humaneval can be non-saturating, it only reflects the model's ability to generate code. A dedicated benchmark for code QA is in demand.
>
> I hope this response motivates InfiBench more clearly. We will add this discussion to our submission.
>
> > Concern 2: Lacking comparison with human judgment.
>
> Thank you very much for the detailed suggestion! We follow your suggestion and conduct additional human evaluation experiments to compare the InfiBench evaluation and model(GPT-4o)-based evaluation on 100 randomly sampled model response pairs. **The detailed experimental setup, results, and implications are in the top general rebuttal post**. In summary,
>
> - for the model response pairs where both human and InfiBench have a clear preference, their agreement rate is **85.1%**;
>
> - for the model response pairs where both human and GPT-4o have a clear preference, their agreement rate is **77.8%**.
>
> As a result, the InfiBench model-free evaluation aligns with human experts better than the GPT-4o evaluation. The advantage of InfiBench comes from the ability to detect deceptive answers. Specifically, some model responses pretend to be helpful with lengthy wording and hallucinations, and InfiBench is stronger at detecting them than GPT-4o by looking for key concepts that should exist in a helpful answer. We further point out that InfiBench evaluation is more efficient and cost-saving than model-based evaluation. On the other hand, the study found that InfiBench evaluation could be too strict sometimes due to pattern matching and fixed post-processing. Detailed discussion is in the top general rebuttal post.
>
> We will add the human evaluation experiment in revision.
>
> > Concern 3: Section 4 is not very insightful. Most findings summarise the experiment results rather than provide insights.
>
> Thanks for the comment. We have more findings and analysis in the appendices, and we will move them to the main text to bring more insights in Section 4. For example:
>
> - **Failure case analysis**: In Appendix E.4, we identify the over-safeguarding problem in CodeLLama70B; in Appendix H, we present the hard cases (level 4 & 5 questions), which usually correspond to code generation with long and domain-specific context or challenging blank-filling questions which models may not see during instruction tuning.
>
> - **Strengths and weaknesses for representative models**: Table 7 in Appendix D is a thorough demonstration by topics, difficulty levels, and instruction types, where we found hard cases for GPT4 are also generally hard for open-source models.
>
> - **Correlation between models' performance on HumanEval and InfiBench**: As outlined in Appendix E.1 and Figure 5, HumanEval and InfiBench performance has a positive correlation for most models. Scaling or instruction tuning improves model performance more significantly on HumanEval but more moderately on InfiBench. The discrepancy (significantly on HumanEval but more moderately on InfiBench) may correspond to different difficulty levels of the two benchmarks. However, some models are overly optimized on HumanEval - we recommend generalizing the code training data beyond simple coding problems to improve the code LLMs.
>
> - **Scaling-law extrapolation** (new result after submission): Figure 4 presents two arrows representing scaling tendencies by model parameters. After the submission, Deepseek-coder-v2 [1] was released, and it is the largest code LLM to our knowledge in an MoE architecture with 236B parameters in total and 21B activated. On InfiBench, Deepseek-coder-v2 achieves **65.49%**, setting the new benchmark for open-source LLMs but still being inferior to GPT-4o. More importantly, the score is within the predicted range of our empirical scaling law.
>
> -----
>
> We will incorporate the above response and experiment into our revision to improve the submission. Thanks again for your insightful review! If there is any unclear point or your concerns remain, please let us know; we are happy to address them.
>
>
>
> *[1] Zhu, Qihao, et al. "DeepSeek-Coder-V2: Breaking the Barrier of Closed-Source Models in Code Intelligence." arXiv preprint arXiv:2406.11931 (2024).*`

---

> > ### Comment · Reviewer_KYrR · 2024-08-28
> >
> > Thanks for your clarification and new results. I would like to keep my score.

---

### Official Review · Reviewer_AyvS · 2024-07-25
**InfiBench: A comprehensive and insightful benchmark for code LLM question-answering capabilities**

**Rating:** 8
**Confidence:** 4

**Review:**

InfiBench tackles an important challenge in evaluating LLMs for real-world coding scenarios, presenting a novel benchmark dataset and evaluation framework for free-form question-answering tasks.
One of the intriguing findings is that for this benchmark, the scaling laws for code models are significantly different from general LLMs. This definitely needs investigation in future works.

However, there are a few areas that could use some improvement:
* The relatively small number of questions (234) might limit the benchmark's statistical power (although popular benchmarks like HumanEval are on the same scale).
* The sample selection criteria are biased towards the most popular languages.
* The complex evaluation criteria (4 different metrics) may complicate future maintenance and integration of the benchmark.

Despite these limitations, the core contribution is valuable and the paper provides crucial insights into the current state and future directions of code LLMs.

**Strengths:**

* The benchmark uses high-quality, real-world programming questions from Stack Overflow, providing a diverse and representative set of challenges for evaluating code LLMs.
* The authors took steps to reduce domain gap and mitigate data contamination by having domain experts paraphrase the original Stack Overflow questions. This helps simulate more realistic interactions with code LLMs and reduces potential memorization advantages.
* The benchmark's design effectively mitigates the impact of potential memorization, as shown by the relatively low performance of directly copying Stack Overflow answers compared to GPT-4's performance.
* The evaluation framework incorporates multiple model-free metrics, enhancing the robustness and fairness of the assessment.
* The comprehensive evaluation of over 100 code LLMs provides valuable insights into the current state of the art.

**Additional Feedback:**

N/A

**Clarity:**

The paper is generally well-written and easy to follow. The authors provide clear explanations of their methodology, benchmark construction, and findings.

**Correctness:**

The claims made in the submission appear to be correct, and the benchmark is constructed in a sound manner. The evaluation methods and experiment design are extensive, as evidenced by the evaluation of over 100 code LLMs and the detailed explanations of the data collection, filtering, and evaluation processes.

**Documentation:**

The authors provide sufficient detail on data collection, organization, and evaluation methods. They have made the dataset and evaluation framework publicly available, which supports reproducibility.

**Ethics:**

There are no significant ethical concerns with this submission. The authors have addressed potential issues related to data privacy and consent by using publicly available Stack Overflow data (which uses the same CC-BY-SA license as the dataset) and paraphrasing questions. They have also taken steps to remove PII from the dataset. The commitment to responsible AI practices is evident in their approach to data biases and personal sensitive information handling.

**Limitations:**

The authors have adequately addressed the main limitations of their work, particularly regarding potential data contamination. They have taken steps to mitigate these issues, such as paraphrasing questions and analyzing the impact of using original Stack Overflow answers. However, they could provide more detailed discussion on the potential biases introduced by their question selection process and the limitations of their automatic evaluation metrics.
The paper acknowledges several important points about data biases:
* Non-standard evaluation: The benchmark uses fixed hyperparameters for all LLM generations, which could lead to misleading comparisons if not used consistently.
* Potential misinterpretation: The benchmark focuses on response correctness, not other aspects like naturalness or fairness.
* Data contamination risks: Future LLMs might be fine-tuned on this dataset, which is hard to prevent given its public nature.

**Opportunities For Improvement:**

* While the benchmark covers a wide range of programming languages, the total number of questions (234) is relatively small. Expanding the dataset could improve statistical significance and coverage of diverse programming scenarios.
* The complexity of the evaluation criteria (4 fundamentally different automatic metrics) could potentially hinder future maintenance and integration of the benchmark into different evaluation frameworks.
* The selection criteria for questions may introduce bias towards more popular or recent programming topics. Future versions of the dataset could use a more balanced selection of questions, including less frequently viewed but potentially important topics.
* The paper could benefit from a more detailed analysis of how InfiBench performance correlates with real-world programming task performance, possibly through user studies (perhaps the same group of developers that created the dataset itself).

**Relation To Prior Work:**

The authors clearly discuss how InfiBench differs from previous contributions, particularly in its focus on free-form question-answering capabilities. They provide a comprehensive comparison with existing benchmarks, highlighting the unique aspects of InfiBench.

**Summary And Contributions:**

InfiBench adds to the suite of code evaluation benchmarks, but addresses a critical gap by focusing on free-form question-answering capabilities of LLMs. The paper's primary contribution is the creation of a diverse, high-quality dataset comprising 234 human-paraphrased Stack Overflow questions and answers spanning 15 programming languages.
The authors describe in detail the dataset construction and filtering process, and a reproducible model-free automatic evaluation framework. To demonstrate the benchmark's utility, they conduct an evaluation of over 100 code and general LLMs, providing valuable insights into model performance and scaling behavior.
The open-source nature of InfiBench and its planned expansion are absolutely essential to the code LLM evaluation landscape.

---

> ### Author Rebuttal · Authors · 2024-08-19
>
> We thank the reviewer for the constructive and insightful comment! We are pleased that the reviewer finds our benchmark to be high-quality, model-free, comprehensive, and robust to memorization! We answer your concerns and questions below.
>
> > Concern 1: Total number of questions is relatively small.
>
> We acknowledge that it is a limitation that the total number of questions is relatively small. It could be viewed as ensuring quality but sacrificing quantity, given the quantity-quality tradeoff under a given human labor budget. As a result, as our complementary user study shows (see answers for Concern 4), even with limited size, our benchmark evaluation results may be more aligned with a human than model-based evaluation.
>
> On the other hand, we are continuously expanding the benchmark to address the limitation, with [40 questions newly added](https://github.com/infi-coder/infibench-evaluation-harness/tree/expand_set), and models are benchmarking on them. Furthermore, the publication of this submission would foster community involvement in benchmark expansion, which we think is one of the best ways to overcome the size limit.
>
> > Concern 2: The selection criteria for questions may introduce bias towards more popular or recent programming topics.
>
> Thanks for pointing out! During the data selection process, we tried to mitigate the bias problem, as demonstrated by the following table.
>
> | Area       | #, Ratio in Initial Seed Set | #, Ratio in InfiBench | Ratio Diff |
> | ---------------- | ----------------------------------------------------------- | -------------------------------------- | ---------- |
> | Front-End    | 5599, 40.4%                         | 63, 26.9%               | -13.5%   |
> | Back-End     | 2592, 18.7%                         | 77, 32.9%               | +14.2%   |
> | DS & ML     | 2963, 21.4%                         | 56, 23.9%               | +2.5%   |
> | Mobile & Desktop | 2512, 18.1%                         | 19, 8.1%                | -10.0%   |
> | IT Operations   | 188, 1.4%                          | 19, 8.1%                | +6.7%   |
>
> In the table, "#, Ratio" stands for number of samples and ratio. As shown, the original high-quality StackOverflow posts are biased toward front-end, mobile, and desktop topics, and our re-sampling increases the quota for back-end, DS, ML, and IT operations. However, we admit that the number of questions for some less popular programming languages, such as Ruby and Rust, could be bigger. In the next version, we will put a high weight on less frequent but important topics to construct a more balanced benchmark.
>
> > Concern 3: Complexity of the evaluation criteria could potentially hinder future maintenance and integration of the benchmark into different evaluation frameworks.
>
> Thanks for the comment. We understand that our model-free evaluation criteria are more complex than some existing benchmarks that usually have a single criterion (execution, BLUE score, or model-based). We view such complexity as a cost for reliable and efficient evaluation reflecting the diversity of the benchmark questions. For code benchmarks with similar levels of sample diversity, such as CodeXGLUE [1], the multi-criteria-based evaluation is also deployed.
> On the other hand, we are working on lowering the barrier of maintenance and integration. For maintenance, we release the detailed schema for the evaluation script [here](https://github.com/infi-coder/infibench-evaluation-harness/blob/main/evaluation_script_schema.md). We are developing web front-end and tutorials. For integration, InfiBench is seamlessly integrated with the latest branch of bigcode-evaluation-harness, the widely-used open-source code LLM evaluation framework, providing a template for future integration. We will also integrate InfiBench with other evaluation frameworks such as openai/evals.
>
> > Concern 4: A more detailed analysis of how InfiBench performance correlates with real-world programming task performance, possibly through user studies.
>
> Great suggestion! According to our human annotators' feedback, the ranking of models generated from InfiBench among the three models, GPT4 > mistral Codestral 22B > GPT3.5, coincides with their personal feelings of helpfulness in their daily software development. To study the correlation with human preference more quantitatively, we conduct additional experiments to compare the InfiBench evaluation and model(GPT-4o)-based evaluation on 100 randomly sampled model response pairs. **The detailed experimental setup, results, and implications are in the top general rebuttal post**. In summary,
>
> - for the model response pairs where both human and InfiBench have a clear preference, their agreement rate is **85.1%**;
>
> - for the model response pairs where both human and GPT-4o have a clear preference, their agreement rate is **77.8%**.
>
> As a result, the InfiBench evaluation aligns with human experts better than the GPT-4o evaluation. The advantage of InfiBench comes from the ability to detect deceptive answers. Specifically, some model responses pretend to be helpful with lengthy wording and hallucinations, and InfiBench is stronger at detecting them than GPT-4o by looking for key concepts that should exist in a helpful answer. On the other hand, the study found that InfiBench evaluation could be too strict sometimes due to pattern matching and fixed post-processing. A detailed discussion is in the top general rebuttal post.
>
> -----
>
> We will incorporate the above response and experiment into our revision to improve the submission. Thanks again for your insightful review! If there is any unclear point or your concerns remain, please let us know; we are happy to address them.
>
>
>
> *[1] Lu, Shuai, et al. "CodeXGLUE: A Machine Learning Benchmark Dataset for Code Understanding and Generation." NeurIPS 2021 Datasets and Benchmarks Track.*

---

> > ### Author Response · Authors · 2024-08-29
> > **A Kind Reminder of Paper Discussion Period**
> >
> > Dear Reviewer AyvS,
> >
> > Thanks again for your appreciation of our work! As the end of the discussion (Aug 31 AoE) is approaching, we really appreciate it if you can consider our responses to your questions. We are very thankful for your comments and suggestions that helped improve our paper, and we will incorporate them in our revision. We also posted new results in the top rebuttal post. Please let us know if you have any questions, and we are more than happy to address them. Thanks!

---

> > ### Comment · Reviewer_AyvS · 2024-08-29
> >
> > Thanks a lot for the additional results and addressing all of the concerns. Keeping the score at 8.

---

### Official Review · Reviewer_meDx · 2024-07-30
**The paper introduces InfiBench, a question-answering benchmark for code**

**Rating:** 4
**Confidence:** 5
**Correctness:** N/A
**Clarity:** The paper is well-written and easy to…

**Review:**

The paper is well-structured and easy to follow.
Pros:
- The paper proposes a QA benchmark for code, distinguishing itself from the primarily code generation datasets used in previous studies.
- The authors conducted extensive evaluation on over 100 LLMs, which gives valuable and insightful observation for the community.
- The benchmark and evaluation tools are fully open source, promoting transparency and continuous improvement.
Cons:
- Limited Size: The benchmark includes only 234 questions, which may not fully capture the diversity of real-world coding scenarios.
- Evaluation metrics based on match-based methods are weak and may require validation against human evaluations.
- The insights gained from the evaluation lack novelty.

**Strengths:**

- The paper introduces InfiBench, a benchmark for evaluating the question-answering capabilities of code LLMs, addressing a gap in existing benchmarks.
- InfiBench spans 15 programming languages and 5 major areas, making it highly relevant for diverse real-world coding domains.
- The benchmark is created from high-quality Stack Overflow questions and includes a systematic evaluation of over 100 code LLMs, providing insightful findings.
- The benchmark is open source and systematic practices in code LLM evaluation, with measures to mitigate data contamination and ensure privacy.

**Additional Feedback:**

N/A

**Documentation:**

Yes

**Limitations:**

The authors need to conduct a more thorough review of the literature. For instance, the statement “Benchmarks beyond code generation are relatively few” is inaccurate. Benchmarks for code intelligence tasks have been around for some time, such as CodeXGLUE (https://github.com/microsoft/CodeXGLUE), which includes various tasks related to software engineering, not just code generation. The authors should also compare their work with CodeXGLUE.

Additionally, the benchmark used in the study is too small, comprising only ~200 samples. This limited size cannot provide a comprehensive evaluation for software engineering tasks as claimed. The authors should expand the benchmark size before making such a claim.

**Opportunities For Improvement:**

- The benchmark's limited size results in some topics within InfiBench having only a few samples (e.g., Mobile & Desktop and IT operations each have only 19 samples). This limitation may lead to biased results and less robust evaluations.
- The paper does not mention other QA datasets for code such as [CodeQA](https://arxiv.org/abs/2109.08365) for comparison.
- Match-based metrics are only partially reliable for evaluating freeform QA tasks. The experiment should conduct human evaluation or model-based evaluation or showing the high reliability of these metrics.
- Some insights are not clearly discussed. The experiment mainly focuses on model performance without an in-depth discussion. For instance, it does not thoroughly examine failed examples across models, the strengths and weaknesses of each model in different topics and languages, or why instruction models improve performance on HumanEval but decline on InfiBench.
- Provide suggestions or proposed techniques for improving existing models on this QA task for code.

**Relation To Prior Work:**

Not clearly describe the literature, need to expand.

**Summary And Contributions:**

The paper introduces InfiBench, a question-answering benchmark for code, comprising 234 questions sourced from Stack Overflow across 15 programming languages and 5 topics. It uses four automatic metrics for evaluation: keyword matching, blank filling, unit testing, and dialogue similarity. The study evaluates a substantial number of models, encompassing over 100 LLMs.

---

> ### Author Rebuttal · Authors · 2024-08-19
>
> We thank the reviewer for the constructive and insightful comment! We answer your concerns and questions below.
>
> > Concern 1: Benchmark is limited in size, and cannot provide a comprehensive evaluation for software engineering tasks, with some topics having only a few samples.
>
> We acknowledge that with 234 questions, the benchmark may be limited in size. However, we remark that:
>
> (1) Expert-level annotation, which is inevitable for a reliable free-form QA evaluation, requires substantial human labor costs, especially compared to other test-case-based, match-score-based, or model-based benchmarks. Hence, for the challenging free-form QA evaluation task, **we believe that the publication of this submission would foster community involvement in benchmark expansion, which might be one of the best ways to overcome the size limit**. Besides, we ourselves are continuously expanding the benchmark, with [40 questions newly added](https://github.com/infi-coder/infibench-evaluation-harness/tree/expand_set), and models are benchmarking on them. Meanwhile, our expansion plan will value your suggestions and increase the sampling ratio for less frequent but important topics.
>
> (2) Existing benchmarks with high-quality annotations and human-based filtering are at a similar scale. For example, HumanEval has 164 problems, and CodeContests has 165 problems, both on a relatively smaller scale but widely used as reliable benchmarks. Notice that the recent [SWE-bench Verified](https://openai.com/index/introducing-swe-bench-verified/) from OpenAI has 500 samples at a similar scale despite involving a large group of annotators.
>
> Hence, although the size is limited, it may be a minor limitation and could be mitigated with our continuous expansion and community involvement once published.
>
> > Concern 2: Evaluation metrics are weak and human evaluation or model-based evaluation is needed to show high reliability.
>
> Thanks for the suggestion! In response to this concern, we conduct additional human experiments to compare the InfiBench evaluation and model(GPT-4o)-based evaluation on 100 randomly sampled model response pairs. **The detailed experimental setup, results, and implications are in the top general rebuttal post**. In summary,
>
> - for the model response pairs where both human and InfiBench have a clear preference, their agreement rate is **85.1%**;
>
> - for the model response pairs where both human and GPT-4o have a clear preference, their agreement rate is **77.8%**.
>
> As a result, the InfiBench model-free evaluation aligns with human experts better than the GPT-4o evaluation. The advantage of InfiBench comes from the ability to detect deceptive answers. Specifically, some model responses pretend to be helpful with lengthy wording and hallucinations, and InfiBench is stronger at detecting them than GPT-4o by looking for key concepts that should exist in a helpful answer. We further point out that InfiBench evaluation is more efficient and cost-saving than model-based or human-based evaluation. We will add the human experiment in revision.
>
> > Concern 3: Some insights are not clearly discussed. Need to provide suggestions or proposed techniques for improving existing models on this QA task for code.
>
> We thank the reviewer for the constructive comment for improving the insight discussion!
>
> - For failure case analysis, in Appendix E.4, we identify the over-safeguarding problem in CodeLLama70B; in Appendix H, we present the hard cases (level 4 & 5 questions), which usually correspond to code generation with long and domain-specific context or challenging blank-filling questions which models may not see during instruction tuning.
>
> - For strengths and weaknesses in each model, Table 7 in Appendix D is a thorough demonstration by topics, difficulty levels, and instruction types. We found hard cases for GPT4 are also generally hard for open-source models.
>
> - For instruction models' performance on HumanEval and InfiBench, as outlined in Appendix E.1 and Figure 5, for most models, we do not find that instruction tuning improves on HumanEval but declines on InfiBench. Actually, scaling or instruction tuning both improves more significantly on HumanEval but more moderately on InfiBench. The discrepancy may correspond to different difficulty levels of the two benchmarks. As a result, we propose to generalize the code training data beyond simple coding problems to improve the code LLMs, and we defer the technique to another report to avoid clutter.
>
> We will move the discussion on these aspects from appendices to the main text and further expand it in revision.
>
> > Concern 4: Conduct a more thorough review of the literature. Benchmarks for code intelligence tasks have been around for some time, such as CodeXGLUE.
>
> Thanks for pointing out CodeXGLUE! CodeXGLUE is one of the few benchmarks beyond code generation. It is a systematic benchmark, i.e., a "meta-benchmark", by integrating several sub-datasets. In contrast, InfiBench is also beyond code generation, but more specifically focusing on the new and unexplored topic of free-form QA, which is a task not covered by existing CodeXGLUE tasks or other code benchmarks to our knowledge. Moreover, InfiBench is featured by the original data source and diverse correctness evaluation criteria. **Instead of viewing CodeXGLUE (or other code benchmarks) as similar or overlapping benchmarks to InfiBench, we believe they are complementary.** For example, CodeXGLUE, as a "meta-benchmark", could be benefited from integrating InfiBench to cover the task of free-form QA. We will add this discussion in revision.
>
> -----
>
> Thanks again for your insightful review! We sincerely hope you read our response. If the response resolves your concerns, your re-evaluation would be greatly appreciated. We are more than happy to discuss this further. Please let us know.
>
>
>
> *[1] Li, Yujia, et al. "Competition-level code generation with alphacode." Science 378.6624 (2022): 1092-1097.*

---

> > ### Author Response · Authors · 2024-08-29
> > **A Kind Reminder of Paper Discussion Period**
> >
> > Dear Reviewer meDx,
> >
> > As the end of the discussion (Aug 31 AoE) is approaching, we really appreciate it if you can consider our responses to your questions when evaluating our work. We are very thankful for your comments and suggestions that helped improve our paper, and we will incorporate them in our revision. We also posted new results in the top rebuttal post. We would greatly appreciate it if you would consider raising your score in light of our response. Please let us know if you have any questions, and we are more than happy to address them. Thanks!

---

### Author Rebuttal · Authors · 2024-08-19

We thank all reviewers for their insightful reviews, which would significantly improve the manuscript. Please find our response to each reviewer's concerns and questions in the corresponding rebuttal post. In this rebuttal post, we present (1) an additional human evaluation as pointed out by $\textcolor{purple}{\textrm{Reviewer meDx}}$, $\textcolor{orange}{\textrm{Reviewer AyvS}}$, and $\textcolor{blue}{\textrm{Reviewer kYrR}}$; and (2) recent updates to InfiBench, including new benchmark questions and new evaluation results. We will integrate both results into the manuscript revision.

## Human Evaluation Experiment

Our human evaluation compares the alignment of automatic evaluation results with human expert evaluation. We consider two automatic evaluation methods: InfiBench and GPT-4o.

### Data Generation

We randomly sample 100 questions without replacement from InfiBench. We select three strong LLMs to construct response pairs: GPT-4-0613, GPT-3.5-turbo, and Mistral Codestral 22B. For each question, we randomly choose two out of these three models, generate responses, and label them as Response A / Response B, respectively.

### Response Evaluation

**For InfiBench evaluation**, we feed in responses to grade. Recall that the InfiBench score is within the range of 0.0 - 1.0 for each response, and the concrete evaluation criteria vary for different questions. For each response pair, we label four outcomes: If the score difference is larger than 0.2, we label the outcome to be A better than B (A>B) or B better than A (B>A); otherwise, if the maximum score among the two is larger than 0.5, we label the outcome to be A and B both good (A=B↑); otherwise, we label the outcome to be A and B both bad (A=B↓).

**For GPT-4o evaluation**, we deploy the prompting template from LLM-Blender ([1, Appendix E]) and trigger GPT-4o for grading the four outcomes: A>B, B>A, A=B↑, and A=B↓. We enhance the reliability of the comparison by switching A and B and triggering the model twice. We record the preference only when a consistent preference exists.

**For human evaluation**, we recruit the same human annotators to label the comparison preference since they are familiar with the questions and have strong expertise. To ensure fairness, the annotator has no access to the evaluation results of InfiBench and GPT-4o, nor which source model generates the response. Same as InfiBench and GPT-4o evaluation, human annotators are instructed to label each pair with the four outcomes.

### Experiment Result

Here is the consensus matrix between InfiBench and human annotators.

| Number of Pairs | Human: A>B | Human: B>A | Human: A=B↑ | Human A=B↓ | Tot  |
| --------------- | ---------- | ---------- | ----------- | ---------- | ---- |
| **InfiBench: A>B**  | **23**     | **3**      | 9           | 4          | 39   |
| **InfiBench: B>A**  | **4**      | **17**     | 12          | 2          | 35   |
| **InfiBench: A=B↑** | 0          | 0          | 10          | 0          | 10   |
| **InfiBench: A=B↓** | 4          | 3          | 3           | 6          | 16   |
| **Tot**             | 31         | 23         | 34          | 12         | 100  |

If we only count the cases where both human and InfiBench have clear preferences, their agreement rate is (40 / 47) = **85.1%**.

| Number of Pairs | Human: A>B | Human: B>A | Human: A=B↑ | Human A=B↓ | Tot  |
| ------------ | ---------- | ---------- | ----------- | ---------- | ---- |
| **GPT-4o: A>B**  | **23**     | **7**      | 8           | 6          | 44   |
| **GPT-4o: B>A**  | **3**      | **12**     | 9           | 3          | 27   |
| **GPT-4o: A=B↑** | 5          | 4          | 15          | 3          | 27   |
| **GPT-4o: A=B↓** | 0          | 0          | 2           | 0          | 2    |
| **Tot**          | 31         | 23         | 34          | 12         | 100  |

If we only count the cases where both human and GPT-4o have clear preferences, their agreement rate is (35 / 45) = **77.8%**.

### Implications

1. From the agreement rate comparison, the InfiBench evaluation aligns with human experts better than the GPT-4o evaluation (with >80% confidence). Due to the time limit, we picked only 100 cases for study and will further enlarge the study scale to improve confidence level. The advantage of InfiBench comes from the ability to detect deceptive answers. Specifically, some model responses pretend to be helpful with lengthy wording and hallucinations, and InfiBench is stronger at detecting them than GPT-4o by looking for key concepts that should exist in a helpful answer.

2. From the human-GPT-4o matrix, we identify that GPT-4o falls short when A and B are both bad responses, labeling none of them as "both bad".

3. From the InfiBench-human matrix, we identify that InfiBench evaluation could be too strict due to pattern matching and fixed post-processing leading to over-differentiation - when human believes A and B are both good responses, with only a 29.4% chance InfiBench labels them as "both good".

The raw experimental data is available [here](https://github.com/infi-coder/infibench-evaluation-harness/tree/main/bigcode_eval/infibench/human_study).

## Recent InfiBench Updates

Our benchmark is continuously expanding:

1. After the submission, we have added [40 new cases](https://github.com/infi-coder/infibench-evaluation-harness/tree/expand_set), and models are benchmarking on them.

2. After the submission, we benchmarked two new strong LLMs: Deepseek-coder-v2 achieves a score of 65.49%, setting the new state-of-the-art among open-source models and being second only to GPT4. Claude 3.5 Sonnet achieves a score of 58.59%, being almost on par with Claude 3 Sonnet (58.20%) but inferior to Claude 3 Opus (63.89%). We welcome any suggestions on more model evaluation.



*[1] Jiang, Dongfu, Xiang Ren, and Bill Yuchen Lin. "LLM-Blender: Ensembling Large Language Models with Pairwise Ranking and Generative Fusion." ACL 2023*

---

### Comment · Area_Chair_Rjia · 2024-08-28
**Important steps before conclusion of discussion period**

Dear Reviewers, thank you for your constructive comments. If you have not done so already, please review and respond to author rebuttals to your review. At a minimum, please acknowledge that you have received the rebuttal, indicate whether/why the points you raised in your original review have/have not been addressed, and provide some reasoning.

If you have already acknowledged author rebuttals, any further engagement is at your discretion.

The discussion period is set to end this Saturday, Aug. 31, and for our purposes the deadline will be Anywhere on Earth.

---

### Decision · Program_Chairs · 2024-09-26

**Decision:**

Accept (Poster)

**Comment:**

This paper contributes a dataset that is novel for its focus on freeform code question-answering as opposed to code generation.

Reviewers (meDx, KYrR) observe that the authors perhaps go too far in their assertion that the dataset is unique as opposed to a meaningful contribution in a less-explored area.

Authors address a variety of reviewer concerns, chief among them the need for "gold-standard" human evaluation to compare against.

Overall the paper constitutes a strong marginal contribution to an area of research that is now attracting a great deal of interest. As a tightly focused incremental advance in a field of great interest --- that, importantly, receives high praise even from reviewers that do not believe it should be accepted for oral presentation --- it is a strong candidate for a poster presentation.